# FAIRRET: A FRAMEWORK FOR DIFFERENTIABLE FAIRNESS REGULARIZATION TERMS

**Maarten Buyl**
Ghent University
maarten.buyl@ugent.be

**MaryBeth Defrance**
Ghent University
marybeth.defrance@ugent.be

**Tijl De Bie**
Ghent University
tijl.debie@ugent.be

## ABSTRACT

Current fairness toolkits in machine learning only admit a limited range of fairness definitions and have seen little integration with automatic differentiation libraries, despite the central role these libraries play in modern machine learning pipelines.

We introduce a framework of fairness regularization terms (FAIRRETs) which quantify bias as modular, flexible objectives that are easily integrated in automatic differentiation pipelines. By employing a general definition of fairness in terms of linear-fractional statistics, a wide class of FAIRRETs can be computed efficiently. Experiments show the behavior of their gradients and their utility in enforcing fairness with minimal loss of predictive power compared to baselines. Our contribution includes a PyTorch implementation of the FAIRRET framework.

## 1 INTRODUCTION

Many machine learning *fairness* methods aim to enforce mathematical formalizations of non-discrimination principles (Mehrabi et al., 2021), often by requiring statistics to be equal between groups (Agarwal et al., 2018). For example, we may require that men and women receive positive decisions at equal rates in binary classification (Dwork et al., 2012). The main interest in fairness tools is to meet such constraints without destroying the accuracy of the ML model.

A large class of these fairness tools utilizes *regularization terms*, i.e. quantifications of unfairness that can be added to the existing error term of an unfair ML model (Kamishima et al., 2012; Berk et al., 2017; Zafar et al., 2019; Padala & Gujar, 2021; Padh et al., 2021; Buyl & De Bie, 2022). The modularity of such loss terms appears to align well with the paradigm of automatic differentiation libraries like PyTorch (Paszke et al., 2019), which have become the bedrock of modern machine learning pipelines. However, the practical use of this modularity has seen little interest thus far.

**Contributions** Hence, we formalize a framework of fairness regularization terms (**FAIRRETs**) that consolidates research in differentiable fairness methods. A FAIRRET quantifies a model's unfairness as a single value that is minimized like any other objective through automatic differentiation.

We implement two types of FAIRRETs: FAIRRETs that directly penalize the *violation* of fairness constraints and FAIRRETs that minimize the distance between a model and its *projection* onto the set of fair models. These FAIRRETs support any fairness notion defined through linear-fractional statistics (Celis et al., 2019), which is a far wider range than the exclusively linear statistics typically considered in literature (Zafar et al., 2019; Agarwal et al., 2018). Moreover, our framework generalizes to the simultaneous handling of multiple sensitive traits and (a weaker form of) fairness with respect to continuous sensitive variables. By design, FAIRRETs are both modular and extensible such that future work that can benefit from their wide applicability. Appendix E contains a code example.

We visualize the FAIRRETs' gradients and evaluate their empirical performance in enforcing fairness notions compared to baselines. We infer this is far more difficult for fairness notions with linear-fractional statistics, which were rarely studied in prior work, than those with linear statistics.

The framework is available as a package at `https://github.com/aida-ugent/fairret`.

**Related Work** Fairness tools are classified as *preprocessing*, *inprocessing* or *postprocessing* (Mehrabi et al., 2021). FAIRRETs perform inprocessing, as they are minimized during training.

A popular approach to fairness regularization is to penalize the violation of fairness constraints (Zemel et al., 2013; Padala & Gujar, 2021; Wick et al., 2019), which we formalize as a FAIRRET. We also take inspiration from postprocessing methods that project classifiers onto a fair set (Alghamdi et al., 2020; Wei et al., 2020) and penalize the cost of this projection (Buyl & De Bie, 2021) as a FAIRRET. Fair representation learning (McNamara et al., 2019; Oneto et al., 2020; Franco et al., 2022) finds intermediate representations that minimally contain sensitive information. An example is the adversarial approach of Adel et al. (2019), which is a baseline in our experiments.

Celis et al. (2019) observed that many fairness definitions express a parity between linear-fractional statistics. They propose a meta-algorithm to find optimal classifiers that satisfy this constraint. Instead, we employ a simpler (yet sufficiently expressive) linear-fractional form and propose an algorithm to use them in the construction of linear constraints that does not require a meta-algorithm.

Popular fairness toolkits such as Fairlearn (Bird et al., 2020) and AIF360 (Bellamy et al., 2018) expect the underlying model in the form of *scikit-learn Estimators*[1] that can be retrained at-will in fairness meta-algorithms. Instead, our proposed FAIRRETs act as a loss term that can simply be added *within* a training step. The aforementioned toolkits have some integration with automatic differentiation libraries in adversarial fairness approaches (Zhang et al., 2018), yet these still require full control over the training process and lack generality in the fairness notions they can enforce.

Two PyTorch-specific projects with similar goals as our paper are FairTorch (Masashi, 2020) and the Fair Fairness Benchmark (FFB) (Han et al., 2023). However, neither present a formal framework and both only support a limited range of fairness definitions.

## 2 FAIRNESS IN BINARY CLASSIFICATION

In fair binary classification, we are provided with random variables $(\mathbf{X}, \mathbf{S}, Y)$ with $\mathbf{X} \in \mathbb{R}^{d_x}$ the feature vector of an individual, $\mathbf{S} \in \mathbb{R}^{d_s}$ their *sensitive* feature vector and $Y \in \{0, 1\}$ the binary output label. In what remains, all expectations are taken over the joint distribution of $(\mathbf{X}, \mathbf{S}, Y)$.

The goal is to learn a classifier $f$ such that its predictions $f(\mathbf{X})$ match $Y$ while avoiding discrimination with respect to $\mathbf{S}$. In this section, we will assume $f$ directly provides binary decisions, i.e. $f : \mathbb{R}^{d_x} \to \{0, 1\}$, as this is expected in traditional formalizations of fairness. However, since such 'hard' classifiers are not differentiable, we will instead be learning probabilistic classifiers in Sec. 3.

Further note that our definition of sensitive features $\mathbf{S}$ as real-valued and $d_s$-dimensional vectors is a generalization of typical fairness definitions which assume a categorical (or binary) domain for sensitive features (Verma & Rubin, 2018). We will one-hot encode such categorical traits, e.g. by encoding 'white' or 'non-white' as the vectors $\mathbf{S} = (1, 0)^\top$ and $\mathbf{S} = (0, 1)^\top$ respectively. Our generalization allows us to take multiple non-exclusive sensitive traits into account by mapping them to different values $S_k$ in the same vector $\mathbf{S}$ for $k \in [d_s] = \{0, ..., d_s - 1\}$. Additionally, by letting $S_k \in \mathbb{R}$, we allow soft specifications of identity rather than requiring hard discretization.

### 2.1 PARTITION FAIRNESS

Though we will allow any feature vector $\mathbf{S} \in \mathbb{R}^{d_s}$ in our framework, popular fairness definitions require every person to belong to exactly one demographic group. We call this *partition fairness*.

**Definition 1.** *In **partition fairness**, $\mathbf{S}$ is a one-hot encoding, i.e. $S_k \in \{0, 1\}$ and $\sum_{k \in [d_s]} S_k = 1$.*

**Example 1.** *A straightforward, popular definition in partition fairness is Demographic Parity (DP), also known as statistical parity (Dwork et al., 2012; Verma & Rubin, 2018). It enforces*

$$\forall k \in [d_s] : P(f(\mathbf{X}) = 1 \mid S_k = 1) = P(f(\mathbf{X}) = 1) \tag{1}$$

*which states that all groups ought to get positive predictions at the same rate (i.e. the overall rate).*

*Let $\gamma(k; f) \triangleq \frac{\mathbb{E}[S_k f(\mathbf{X})]}{\mathbb{E}[S_k]}$. It is easily shown that $\gamma(k; f) = P(f(\mathbf{X}) = 1 \mid S_k = 1)$. Thus also*

$$P(f(\mathbf{X}) = 1 \mid S_k = 1) = P(f(\mathbf{X}) = 1) \iff \gamma(k; f) = \mathbb{E}[f(\mathbf{X})]. \tag{2}$$

---

[1]`https://scikit-learn.org/1.3/developers/develop.html` describes these *Estimators*

Table 1: Fairness definitions and their $\alpha$ and $\beta$ functions. Conditional Demographic Parity encompasses many notions with an arbitrary function $\zeta$ conditioned on the input $\mathbf{X}$.

| Fairness Definition | $\alpha_0$ | $\beta_0$ | $\alpha_1$ | $\beta_1$ |
|---|---|---|---|---|
| Demographic Parity (Dwork et al., 2012) | 0 | 1 | 1 | 0 |
| Conditional Demographic Parity (Wachter et al., 2020) | 0 | $\zeta(\mathbf{X})$ | $\zeta(\mathbf{X})$ | 0 |
| Equal Opportunity (Hardt et al., 2016) | 0 | Y | Y | 0 |
| False Positive Parity (Hardt et al., 2016) | 0 | 1 - Y | 1 - Y | 0 |
| Predictive Parity (Chouldechova, 2017) | 0 | Y | 0 | 1 |
| False Omission Parity | Y | -Y | 1 | -1 |
| Accuracy Equality (Berk et al., 2021) | 1 - Y | 2Y - 1 | 1 | 0 |
| Treatment Equality (Berk et al., 2021) | Y | -Y | 0 | 1 - Y |

In Example 1, fairness is formalized by requiring a statistic $\gamma$ to be equal across groups. This principle can be generalized to a wide class of parity-based fairness notions. In particular, we consider those expressed through *linear-fractional* statistics (Celis et al., 2019).

**Definition 2.** *A **linear-fractional** statistic $\gamma$ computes values $\gamma(k; f) \in \mathbb{R}$ for sensitive variable $S_k$ and classifier $f : \mathbb{R}^{d_x} \to \{0, 1\}$. We assume $\gamma$ is differentiable with respect to $f$. It takes the form*

$$\gamma(k; f) = \frac{\mathbb{E}[S_k(\alpha_0(\mathbf{X}, Y) + f(\mathbf{X})\beta_0(\mathbf{X}, Y))]}{\mathbb{E}[S_k(\alpha_1(\mathbf{X}, Y) + f(\mathbf{X})\beta_1(\mathbf{X}, Y))]} \tag{3}$$

*with $\alpha_0$, $\alpha_1$, $\beta_0$, and $\beta_1$ all functions that do not depend on $\mathbf{S}$ or $f$. Let $\Gamma$ denote all such statistics. Also, let $\overline{\gamma}(f) \triangleq \frac{\mathbb{E}[\alpha_0(\mathbf{X}, Y) + f(\mathbf{X})\beta_0(\mathbf{X}, Y)]}{\mathbb{E}[\alpha_1(\mathbf{X}, Y) + f(\mathbf{X})\beta_1(\mathbf{X}, Y)]}$ denote the overall statistic value without conditioning on $\mathbf{S}$.*

**Definition 3.** *A **fairness notion** is expressed through a statistic $\gamma \in \Gamma$. The set $\mathcal{F}_\gamma$ of classifiers that adhere to the fairness notion is defined as*

$$\mathcal{F}_\gamma \triangleq \left\{ f : \mathbb{R}^{d_x} \to \{0, 1\} \mid \forall k \in [d_s] : \gamma(k; f) = \overline{\gamma}(f) \right\} \tag{4}$$

*i.e. the statistic $\gamma(k; f)$ for each $S_k$ equals the overall statistic $\overline{\gamma}(f)$.*

Indeed, the DP fairness notion in Example 1 is expressed as a fairness notion as defined in Def. 3 with linear-fractional statistics as defined in Def. 2. The same holds for the following notions.

**Example 2.** *Equalized Opportunity (EO) (Hardt et al., 2016) only computes DP for actual positives $Y = 1$. Its statistic $\gamma$ is thus the recall $P(f(\mathbf{X}) = 1 \mid Y = 1, S_k = 1)$, i.e. $\gamma(k; f) = \frac{\mathbb{E}[S_k f(\mathbf{X})Y]}{\mathbb{E}[S_k Y]}$.*

**Example 3.** *Predictive Parity (PP) (Chouldechova, 2017), which compares the precision statistic $P(Y = 1 \mid f(\mathbf{X}) = 1, S_k = 1)$, i.e. $\gamma(k; f) = \frac{\mathbb{E}[S_k f(\mathbf{X})Y]}{\mathbb{E}[S_k f(\mathbf{X})]}$.*

**Example 4.** *Treatment Equality (TE) (Berk et al., 2021) balances the ratios of false negatives over false positives, i.e. $\gamma(k; f) = \frac{\mathbb{E}[S_k(1-f(\mathbf{X}))Y]}{\mathbb{E}[S_k f(\mathbf{X})(1-Y)]}$. Unlike the other notions, its $\gamma$ is not a probability.*

Table 1 summarizes the $\alpha$ and $\beta$ functions of several fairness notions (Verma & Rubin, 2018) with linear-fractional statistics. Their derivations are found in Appendix A.1.

**Definition 4.** *A linear-fractional statistic $\gamma \in \Gamma$ is **linear** when $\beta_1(\mathbf{X}, Y) \equiv 0$.*

*Let $\Gamma_L \subset \Gamma$ denote the set of all linear statistics.*

Fairness notions with linear statistics $\gamma \in \Gamma_L$ are thus identified in Table 1 by checking the column for $\beta_1$. Such notions are especially useful because the fairness constraint in Def. 3 is easily written as a linear constraint over classifier $f$. In turn, this makes the set of fair classifiers $\mathcal{F}_\gamma$ a convex set, which leads to convex optimization problems (Boyd & Vandenberghe, 2004). Thus, the constrained optimization of $f$ can be efficiently performed if $f$ is itself linear (Zafar et al., 2019).

However, fairness notions with linear-fractional statistics $\gamma \in \Gamma \setminus \Gamma_L$ do not directly lead to linear constraints in Def. 3. To facilitate optimization, we therefore propose to narrow the set of fair classifiers $\mathcal{F}_\gamma$ to the subset where the statistics are all equal in a particular value $c$.

**Definition 5.** *Fix a $c \in \mathbb{R}$. A **c-fixed fairness notion** is expressed through a linear-fractional statistic $\gamma \in \Gamma$ such that the set $\mathcal{F}_\gamma(c)$ of classifiers $f$ that adhere to the fairness notion is defined as*

$$\mathcal{F}_\gamma(c) \triangleq \left\{ f : \mathbb{R}^{d_x} \to \{0, 1\} \mid \forall k \in [d_s] : \gamma(k; f) = c \right\}. \tag{5}$$

**Proposition 1.** *With $\gamma \in \Gamma$, the c-fixed fairness notion $\mathcal{F}_\gamma(c)$ enforces **linear** constraints:*

$$\gamma(k; f) = c \iff \mathbb{E}[S_k(\alpha(\mathbf{X}, Y, c) + f(\mathbf{X})\beta(\mathbf{X}, Y, c))] = 0 \tag{6}$$

*where $\alpha(\mathbf{X}, Y, c) = \alpha_0(\mathbf{X}, Y) - c\alpha_1(\mathbf{X}, Y)$ and $\beta(\mathbf{X}, Y, c) = \beta_0(\mathbf{X}, Y) - c\beta_1(\mathbf{X}, Y)$.*

Using Prop. 1, we can still obtain linear constraints for fairness notions $\mathcal{F}_\gamma$ with linear-fractional statistics $\gamma \in \Gamma \setminus \Gamma_L$ by considering their $c$-fixed variant $\mathcal{F}_\gamma(c)$ instead. This sacrifices a degree of freedom because statistics $\gamma(k; f)$ are no longer allowed to be equal for any overall statistic $\overline{\gamma}(f)$, they must now do so for the specific case where $\overline{\gamma}(f) = c$. However, there are $c$ values that still lead to interesting sets $\mathcal{F}_\gamma(c)$. In the FAIRRETs we propose, we take an unfair classifier $h$ and fix $c = \overline{\gamma}(h)$ to construct the set of all *fair* classifiers $\mathcal{F}_\gamma(\overline{\gamma}(h))$ that would result from a fair redistribution of scores in $h$ over the sensitive groups.

Though Prop. 1 is inspired by Celis et al. (2019), our use of this result vastly differs. Instead of fixing the statistics to a single value $c$, they set many pairs of upper and lower bounds for each group's statistics, giving rise to as many optimization programs. They then propose a meta-algorithm that searches the best classifier over each of these programs. A meta-algorithm is not necessary in our framework, as we will allow $c$ to evolve during training. While we have no formal convergence guarantees for this approach, empirical results show it works well in practice.

## 2.2 BEYOND PARTITION FAIRNESS

Having firmly rooted our definitions in partition fairness (Def. 1), we now abandon its assumptions. First, we allow $S_k \in \mathbb{R}$. Second, we extend to multiple sensitive features with $\sum_k S_k \in \mathbb{R}$.

### 2.2.1 CONTINUOUS SENSITIVE VALUES

Admitting continuous values for someone's sensitive trait, i.e. $S_k \in \mathbb{R}$ allows us to take naturally continuous features, such as age, into account. Also, it provides an opportunity for an imprecise specification of demographic group membership.

For instance, instead of exactly knowing the gender of an individual, we may only have a probability available, e.g. because it is noisily predicted by a third-party classifier, or to protect the individual's privacy. By allowing $S_k \in (0, 1)$, the attribute $S_k$ could then express 'woman-ness' instead of a binary 'woman' or 'not woman'. Thus, we also allow individuals to themselves quantify how strongly they identify with a group, rather than requiring a binary membership.

Our notation already generalizes to non-binary $S_k$ values; they can simply be filled in for linear-fractional statistics $\gamma \in \Gamma$ as defined in Def. 2. Fairness as formalized in Def. 3 can then still be enforced through $\gamma(k; f) = \overline{\gamma}(f)$.

**Remark 1.** *Partition fairness constraints stem from the principle of treating distinct groups equally. This does not directly apply for a non-binary $S_k$. For example, if there is only one, continuous sensitive variable $(S_0) = \mathbf{S}$ such as the age of an individual, then we cannot compare $\gamma(0; f)$ to another group's statistics. Instead, $\gamma(0; f)$ must be compared to a value independent of $\mathbf{S}$.*

*Enforcing $\gamma(k; f) = \overline{\gamma}(f)$ is then a sensible choice, as it satisfies key properties one can expect from a fairness measure. First, the constraint is met when $S_k \equiv s$, i.e. when $S_k$ is a deterministic constant. Second, it holds if $S_k$ has no linear influence on the numerator and denominator of $\gamma$, i.e.*

$$\text{cov}(S_k, \alpha_0(\mathbf{X}, Y) + f(\mathbf{X})\beta_0(\mathbf{X}, Y)) = \text{cov}(S_k, \alpha_1(\mathbf{X}, Y) + f(\mathbf{X})\beta_1(\mathbf{X}, Y)) = 0 \implies \gamma(k; f) = \overline{\gamma}(f).$$

*For a full derivation of this result, we refer to Appendix B.1.*

### 2.2.2 MULTIPLE AXES OF DISCRIMINATION

By allowing $\sum_k S_k \in \mathbb{R}$, we support that $\mathbf{S}$ contains information about people from several sensitive traits, e.g. gender, ethnicity, and religion. Because these each form a possible axis of discrimination, we can 'sum' these sources of discrimination by combining the constraints.

For example, if pairs of sensitive features $(S_0, S_1)$ and $(S_2, S_3)$ each partition the dataset, then fairness requires both $\gamma(0; f) = \gamma(1; f) = \overline{\gamma}(f)$ and $\gamma(2; f) = \gamma(3; f) = \overline{\gamma}(f)$. Combined, these constraints make up the fairness definition in Def. 3. The use of one-hot notations for sensitive values thus already allows us to combine axes of discrimination for categorical sensitive traits.

**Remark 2.** *An important limitation is that we only view fairness separately per axis of discrimination. Outside the partition fairness setting, this means that some intersections of sensitive groups, e.g. 'black woman', will not be represented in the constraints that enforce fairness with respect to 'black' and 'woman' separately (Kearns et al., 2018). A toy example is given in Appendix B.2.*

## 3 FAIRNESS REGULARIZATION TERMS

The popular approach to modern machine learning is to construct pipelines consisting of modular, parameterized components that are differentiable from the objective to the input. We therefore use *probabilistic* classifier models $h : \mathbb{R}^{d_x} \to (0, 1)$ from now on, where decisions are sampled from a Bernoulli distribution with parameter $h(\mathbf{X})$. Let $\mathcal{H}$ denote the hypothesis class of these models.

**Remark 3.** *Fairness statistics $\gamma(k; h)$ over the output of a probabilistic classifier $h$ only approximately verify their respective fairness notions, as these were only defined for hard classifiers with a binary output (Lohaus et al., 2020). In Appendix B.3, we discuss the impact of this approximation and how its fidelity can be traded-off with the quality of the gradient of $\gamma(k; h)$ with respect to $h$.*

In binary classification, we minimize a loss $\mathcal{L}_Y(h)$ over the probabilistic classifier $h$ given output labels $Y$, e.g. the cross-entropy. In *fair* binary classification we additionally pursue $h \in \mathcal{F}_\gamma$:

$$\min_{h \in \mathcal{F}_\gamma} \mathcal{L}_Y(h). \tag{7}$$

For linear-fractional statistics, the constraint is linear when considering the $c$-fixed variant of $\mathcal{F}_\gamma$ (using Prop. 1). However, for non-convex models $h$, the constrained optimization of $h$ will remain non-convex as well. In the general case, we thus relax $h \in \mathcal{F}_\gamma$ and instead incur a cost to $h \notin \mathcal{F}_\gamma$.

**Definition 6.** *A **fairness regularization term** (FAIRRET) $R_\gamma(h) : \mathcal{H} \to \mathbb{R}_{\geq 0}$ quantifies the unfairness of the model $h \in \mathcal{H}$ with respect to the fairness notion defined through statistic $\gamma$.*

*A FAIRRET is **strict** if it holds that $h \in \mathcal{F}_\gamma \iff R_\gamma(h) = 0$.*

The objective in Eq. (7) is then relaxed as

$$\min_h \mathcal{L}_Y(h) + \lambda R_\gamma(h) \tag{8}$$

with $\lambda$ a hyperparameter. The objective in Eq. (8) is equivalent to Eq. (7) for $\lambda \to \infty$ if $R_\gamma$ is strict.

**Remark 4.** *We call $R_\gamma$ a regularization term, yet its purpose is* not *to reduce model complexity or improve generalization performance, in contrast to traditional regularization in machine learning (Kukačka et al., 2017). Instead, we aim to limit the hypothesis class of $h$ to the set of fair classifiers.*

In what follows, we introduce two archetypes of FAIRRETs: *violation* and *projection*. We visualize $\nabla_h R_\gamma$ for each FAIRRET in Fig. 1 with $\gamma$ the positive rate statistic (thereby enforcing DP).

### 3.1 VIOLATION FAIRRETS

To quantify $h \notin \mathcal{F}_\gamma$, we can start from the *violation* $\mathbf{v}(h)$ of the constraint that defines $\mathcal{F}_\gamma$:

$$\mathbf{v}_k(h) = \left| \frac{\gamma(k; h)}{\overline{\gamma}(h)} - 1 \right| \tag{9}$$

with $\mathbf{v} : \mathcal{H} \to \mathbb{R}^{d_s}$ a vector-valued function with components $\mathbf{v}_k$. Clearly, $\mathbf{v}(h) = \mathbf{0} \iff h \in \mathcal{F}_\gamma$.

Note that $\mathbf{v}(h)$ is normalized[2] by $\overline{\gamma}(h)$ such that a classifier cannot minimize $\mathbf{v}(h)$ by uniformly downscaling its statistics $\gamma$ without reducing relative differences between groups (Celis et al., 2019).

**Definition 7.** *We define the **Norm** FAIRRET as $R_\gamma(h) \triangleq \|\mathbf{v}(h)\|$, with $\|\cdot\|$ a norm over $\mathbb{R}^{d_s}$.*

Many variants of the Norm FAIRRET have been proposed, e.g. by Zemel et al. (2013), Padala & Gujar (2021), Wick et al. (2019) and Chuang & Mroueh (2020). However, fairness evaluation metrics often only consider the maximal violation. Hence, we propose the SmoothMax variant.

---

[2]In cases where $\overline{\gamma}(h) = 0$, we can simply use $\mathbf{v}_k(h) = |\gamma(k; h)|$ instead. We assume $h(\mathbf{X}) \in (0, 1)$, so this only occurs in degenerate cases for the notions in Table 1 (like when all $Y = 0$ for Equal Opportunity).

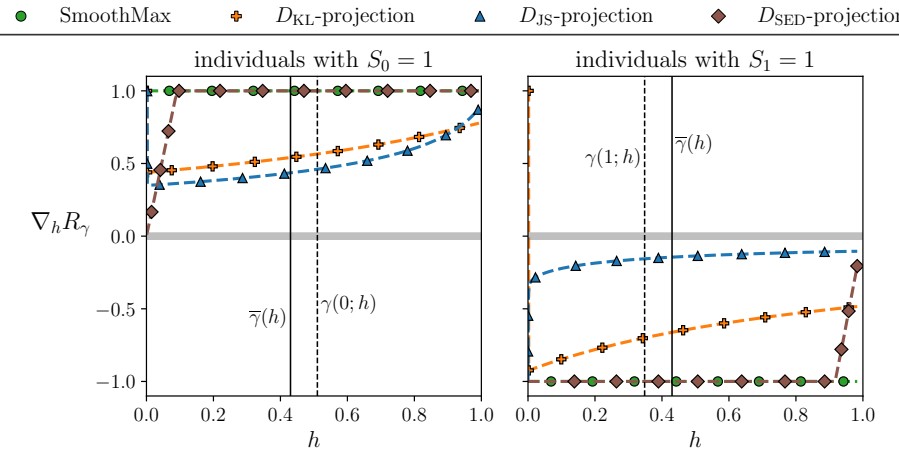

Figure 1: The model $h$ was trained on the *ACSIncome* dataset without FAIRRET (i.e. $\lambda = 0$) and ends up with disparate positive rates $\gamma(0; h) > \overline{\gamma}(h) > \gamma(1; h)$ for the one-hot encoded sensitive variables $(S_0, S_1)$. These should be brought closer to the overall positive rate $\overline{\gamma}(f)$. We show probability scores $h$ and the gradients[3] of several FAIRRETs $R_\gamma$ with respect to $h$. The gradients are normalized by dividing them by their maximum absolute value per FAIRRET and per group. They are positive for samples with $S_0 = 1$, implying their scores should decrease, and vice versa for $S_1 = 1$.

**Definition 8.** *We define the **SmoothMax** FAIRRET as* $R_\gamma(h) \triangleq \log \sum_{k \in [d_s]} \exp(\mathbf{v}_k(h)) - \log d_s$

Because the SmoothMax performs the log-sum-exp operation over the violation, it can be considered a smooth approximation of the maximum. We subtract $\log d_s$ to ensure the FAIRRET is strict.

Generally, violation FAIRRETs can be characterized as functions of the violation $\mathbf{v}(h)$. This lends them interpretability, but it also means that the gradient[3] $\nabla_h R_\gamma$ decomposes as

$$\nabla_h R_\gamma = \left(\frac{\partial \mathbf{v}}{\partial h}\right)^\top \nabla_{\mathbf{v}} R_\gamma \tag{10}$$

with $\frac{\partial \mathbf{v}}{\partial h}$ the Jacobian[3] of $\mathbf{v}(h)$. The gradients of violation FAIRRETs $R_\gamma$ thus only differ in the $\nabla_{\mathbf{v}} R_\gamma$ gradient. Hence, the Norm FAIRRET is excluded from Fig. 1 because its gradients equal those of SmoothMax after normalization. Figure 1 also suggests that violation FAIRRETs convey little information on how each individual $h(\mathbf{X})$ score should be modified. Instead, they merely direct scores to uniformly increase or decrease within each group.

### 3.2 PROJECTION FAIRRETS

Recent postprocessing approaches to fairness redistribute all individual probability scores of a model $h(\mathbf{X})$ to a fair scores vector with a minimal loss in predictive power. For example, Alghamdi et al. (2020) project the scores onto the fair set $\mathcal{F}_\gamma$ as a postprocessing step. Yet, the cost of this projection can be seen as a quantification of unfairness that may be minimized as a FAIRRET during training.

Given a statistical divergence or distance $D$, we can generally define such a *projection* FAIRRET as

$$R_\gamma(h) \triangleq \min_{f \in \mathcal{F}_\gamma(\overline{\gamma}(h))} \mathbb{E}[D(f(\mathbf{X}) \parallel h(\mathbf{X}))]. \tag{11}$$

Importantly, we do not project $h$ onto the general fair set $\mathcal{F}_\gamma$, but on the $c$-fixed subset $\mathcal{F}_\gamma(c)$ with $c = \overline{\gamma}(h)$. The $c$-fixing is done such that the projection only requires linear constraints for linear-fractional statistics (see Prop. 1). A projection FAIRRET (Eq. 11) is then a convex optimization problem if we limit ourselves to a $D$ that is convex with respect to $f$, which is the case for all projections discussed here. In particular, we $c$-fix to the overall statistic $\overline{\gamma}(h)$ of $h$ because this ensures $h$ can always be projected onto itself if it is already fair, as then $h \in \mathcal{F}_\gamma(\overline{\gamma}(h))$.

---

[3]There is some abuse of notation here. When taking the gradient or Jacobian with respect to $h$, we take it with respect to the vector of $n$ outputs of $h$ for a set of $n$ input features sampled from the distribution over $\mathbf{X}$.

**Definition 9.** *The $D_{\text{KL}}$-projection uses the binary Kullback-Leibler divergence*

$$D_{\text{KL}}(f(\mathbf{X}) \parallel h(\mathbf{X})) \triangleq f(\mathbf{X}) \log \frac{f(\mathbf{X})}{h(\mathbf{X})} + (1 - f(\mathbf{X})) \log \frac{1 - f(\mathbf{X})}{1 - h(\mathbf{X})}. \tag{12}$$

The $D_{\text{KL}}$-divergence is both a Csiszar divergence and a Bregman divergence (Amari, 2009). Also, the cross-entropy error minimized in $\mathcal{L}_Y(h)$ equals $D_{\text{KL}}(Y \parallel h(\mathbf{X}))$ up to a constant. The minimization of Eq. (8) thus comes down to simultaneously minimizing $D_{\text{KL}}$ between $h(\mathbf{X})$ and the data $Y$, and between $h(\mathbf{X})$ and the closest $f \in \mathcal{F}_\gamma(\overline{\gamma}(h))$ (Buyl & De Bie, 2021).

**Definition 10.** *The $D_{\text{JS}}$-projection uses the binary Jensen-Shannon divergence.*

$$D_{\text{JS}}(f(\mathbf{X}) \parallel h(\mathbf{X})) \triangleq \frac{1}{2}D_{\text{KL}}(f(\mathbf{X}) \parallel m(\mathbf{X})) + \frac{1}{2}D_{\text{KL}}(h(\mathbf{X}) \parallel m(\mathbf{X})) \tag{13}$$

*with $m(\mathbf{X}) = \frac{1}{2}f(\mathbf{X}) + \frac{1}{2}h(\mathbf{X})$.*

Just like $D_{\text{KL}}$, the $D_{\text{JS}}$-divergence is a Csiszar divergence. However, the $D_{\text{JS}}$-divergence is symmetric with respect to its arguments $f$ and $h$, which is not the case for the $D_{\text{KL}}$-divergence.

**Definition 11.** *The $D_{\text{SED}}$-projection uses the squared Euclidean distance between the two points $(1 - f(\mathbf{X}), f(\mathbf{X}))$ and $(1 - h(\mathbf{X}), h(\mathbf{X}))$:*

$$D_{\text{SED}}(f(\mathbf{X}) \parallel h(\mathbf{X})) \triangleq 2(f(\mathbf{X}) - h(\mathbf{X}))^2. \tag{14}$$

$D_{\text{SED}}$ is a Bregman divergence between the Bernoulli distributions with parameters $f(\mathbf{X})$ and $h(\mathbf{X})$.

In practice, we evaluate projection FAIRRETs $R_\gamma(h)$ in two steps.

$$\text{(i)} \quad f^* = \underset{f \in \mathcal{F}_\gamma(\overline{\gamma}(h))}{\arg \min} \mathbb{E}[D(f(\mathbf{X}) \parallel h(\mathbf{X}))] \tag{15}$$

$$\text{(ii)} \quad R_\gamma(h) = \mathbb{E}[D(f^*(\mathbf{X}) \parallel h(\mathbf{X}))] \tag{16}$$

While keeping $h$ fixed, step (i) computes the overall statistic $\overline{\gamma}(h)$ and then finds the projection $f^*$ through constrained optimization. Subsequently, step (ii) keeps $f^*$ fixed and computes $\mathbb{E}[D(f^*(\mathbf{X}) \parallel h(\mathbf{X}))]$ as a function of $h$, which we use to compute the gradient with respect to $h$. This gradient differs from the actual gradient of the optimization as a function of $h$ in a projection FAIRRET (Eq. 11), because the latter would require us to treat $f^*$ as a function of $h$. However, by treating $f^*$ as fixed instead (without backpropagating through it), we significantly simplify the FAIR-RET's implementation. The optimization in step (i) can then be solved generically using specialized libraries such as `cvxpy` (Agrawal et al., 2018; Diamond & Boyd, 2016). In our experiments, we find that only 10 optimization steps is enough to get a reasonable approximation of the solution. We discuss this approximation and visualize each projection $f^*$ in Appendix C.1 and C.2 respectively.

Figure 1 shows that the gradients of the projection FAIRRETs increase with higher values of $h$. We hypothesize this occurs when $\gamma(k; h) > \overline{\gamma}(h)$ because $\gamma(k; h)$ is more easily decreased by reducing higher $h$ values than lower ones. Conversely, when $\gamma(k; h) < \overline{\gamma}(h)$, there is more to gain from increasing lower $h$ values than higher ones. The sharp bend of the gradients of the $D_{\text{SED}}$-projection is explained in Appendix C.2 through an analysis of the projected distributions.

### 3.3 ANALYSIS

**Proposition 2.** *All FAIRRETs presented in this paper (i.e. Def. 7, 8, 9, 10 and 11) are strict.*

Hence, all proposed FAIRRETs can indeed be properly regarded as quantifications of unfairness.

They are differentiable with respect to $h$. Violation FAIRRETs owe this to the differentiability of $\gamma$ and projection FAIRRETs to the differentiability of $D$. Hence, FAIRRETs are easily implemented with an automatic differentiation library like PyTorch. The computational overhead is unaffected by the complexity of the parameters $\boldsymbol{\theta}$ of $h$, as the gradients $\nabla_{\boldsymbol{\theta}}\mathcal{L}_Y = \left(\frac{\partial h}{\partial \theta}\right)^\top \nabla_h \mathcal{L}_Y$ and $\nabla_{\boldsymbol{\theta}}R_\gamma = \left(\frac{\partial h}{\partial \theta}\right)^\top \nabla_h R_\gamma$ of both loss functions in the joint objective (Eq. 8) share the computation of the Jacobian $\frac{\partial h}{\partial \theta}$.

It is common to minimize $\mathcal{L}_Y$ using mini-batches; the same batches can be used to minimize $R_\gamma$. Indeed, this is done in our experiments. Though this makes FAIRRETs scalable, insufficient batch

sizes will lead to poor approximations of the statistics $\gamma$. Clearly, the mean violation $\mathbf{v}(h)$ in a violation FAIRRET (Eq. 9) computed over mini-batches is not an unbiased estimate of the actual violation over all data. We report the mean SmoothMax loss for increasing batch sizes in Appendix C.3.

# 4 EXPERIMENTS

## 4.1 SETUP

Experiments were conducted on the *Bank* (Moro et al., 2014), *CreditCard* (Yeh & hui Lien, 2009), *LawSchool*[4], and *ACSIncome* (Ding et al., 2021) datasets. Each has multiple sensitive features, including some continuous. The classifier $h$ was a fully connected neural net with hidden layers of sizes [256, 128, 32] followed by a sigmoid and did not take sensitive features $\mathbf{S}$ as input. We trained with all FAIRRETs discussed in Sec. 3 but only report results of Norm, $D_{JS}$-projection and $D_{SED}$-projection in Appendix C.4 to avoid clutter here. The remaining FAIRRETs, SmoothMax and $D_{KL}$-projection, were representative for their archetype. These are compared against three baselines implemented in the Fair Fairness Benchmark (FFB) by Han et al. (2023), as their implementation provides these baselines as loss terms in idiomatic PyTorch. They are *PRemover* (Kamishima et al., 2012), *HSIC* (Pérez-Suay et al., 2017), and *AdvDebias* (Adel et al., 2019) (where the reverse of the adversary's loss is the fairness loss term). In contrast to the FAIRRET implementations, they only accept a single, categorical sensitive attribute. Each FAIRRET and FFB fairness loss was added to the cross-entropy loss in the objective (Eq. 8) in a separate training run for a range of strengths $\lambda > 0$.

We measured fairness over the four statistics $\gamma$ in Table 1 that relate to Demographic Parity (DP), Equal Opportunity (EO), Predictive Parity (PP), and Treatment Equality (TE) respectively. Violation of each fairness notion is computed as $\max_k \mathbf{v}_k(h)$ (see Eq. (9)). Each FAIRRET was minimized with respect to each $\gamma$ in a separate training run (and only the optimized violation is reported). The three FFB baselines only consider one fairness notion, which is to maximize independence between the model's output and the sensitive attributes. Their violation is reported for each statistic $\gamma$.

In summary, there was an experiment run for each dataset, fairness method, fairness strength $\lambda$, and statistic $\gamma$ (except for the FFB baselines). Finally, we also use the *Unfair* baseline with $\lambda = 0$. Each of these combinations was repeated across 10 random seeds with each different train/test splits.

## 4.2 RESULTS

Test set results are visualized in Fig. 2; train set results are found in Appendix C.5 (and display the same trends). We separately discuss the notions with linear and with linear-fractional statistics.

**For DP and EO, which have *linear* statistics**, both the SmoothMax and $D_{KL}$-projection FAIRRETs are effectively used to minimize the fairness violation with respect to multiple sensitive attributes while minimally suffering a loss in AUROC scores, though the projection FAIRRET clearly performs better than the violation-based SmoothMax FAIRRET. As expected, the FFB baselines perform worse than the methods implemented in our FAIRRET framework, since they cannot be configured to optimize the same general range of fairness definitions. Also, their implementation only minimizes bias with respect to a single sensitive attribute, and so they are oblivious to some of the components in $\mathbf{S}$ that the violation in Fig. 2 measures. We report their violations on this single attribute in Appendix C.6, though the FAIRRETs still outperform them there as well.

**For PP and TE, which have *linear-fractional statistics***, all methods appear to struggle far more. SmoothMax is most consistent and never makes the fairness violation worse, yet the $D_{KL}$-projection in most cases makes both the fairness violation and the AUROC worse. The same occurs for the FFB baselines. To some extent, this can be attributed to overfitting, as SmoothMax leads to a significantly more consistent reduction of the train set fairness violation than the test set (see Appendix C.5). Still, non-linear fairness notions are clearly harder to optimize, which aligns with the results of Celis et al. (2019). Though Barocas et al. (2019) conclude that sufficiency (a notion related to PP) 'often comes for free', further work is needed to better understand how such notions can be consistently achieved.

---

[4]Curated and published by the SEAPHE project

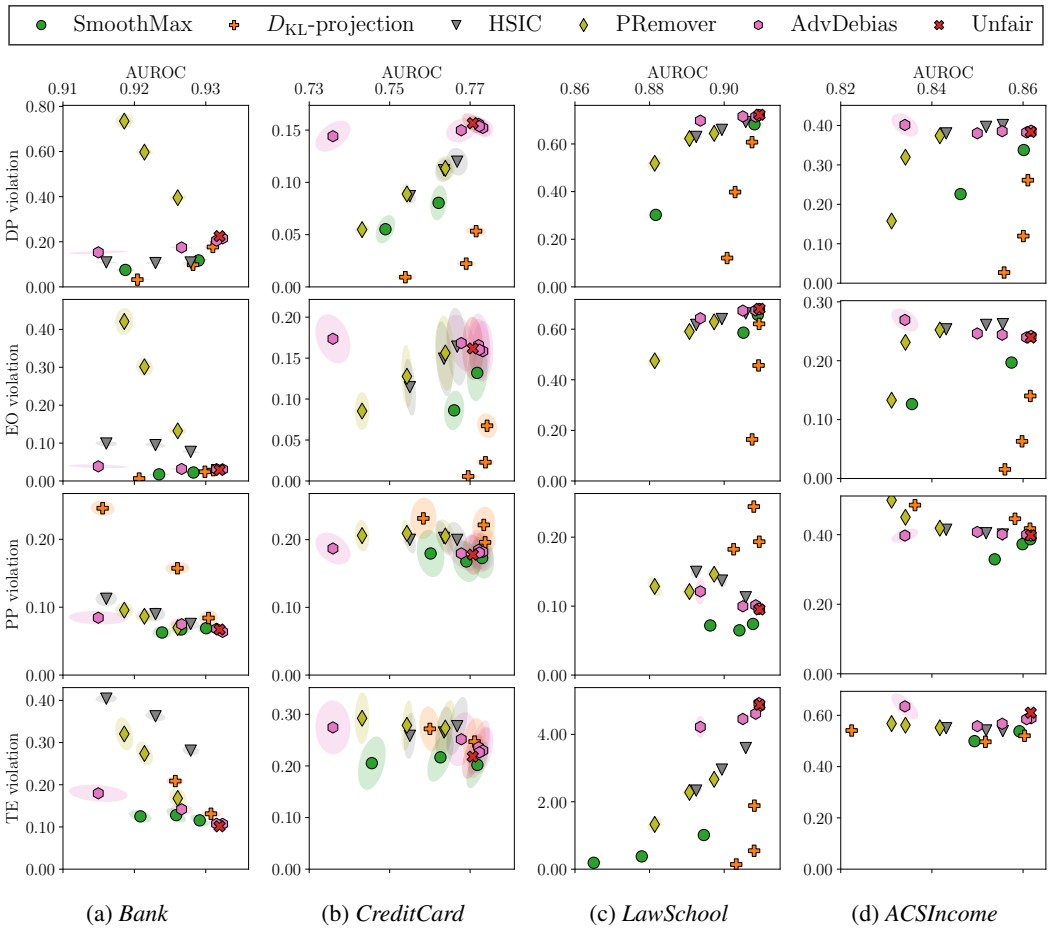

Figure 2: Mean test set results with confidence ellipse for the standard error. Each marker is a separate combination of dataset, FAIRRET, FAIRRET strength, and statistic. Results in the lower right are optimal. Failed runs (with an AUROC far worse than the rest) are omitted.

## 5 CONCLUSION

The FAIRRET framework allows for a wide range of fairness definitions by comparing linear-fractional statistics for each sensitive feature. We implement several FAIRRETs and show how they are easily integrated in existing machine learning pipelines utilizing automatic differentiation.

Empirically, violation FAIRRETs like SmoothMax consistently lead to trade-offs between fairness and AUROC, though the more involved projection FAIRRETs like the $D_{KL}$-projection clearly outperform them on fairness definitions with linear statistics. However, all methods struggle with fairness notions that have linear-fractional statistics like PP and TE, which have mostly been ignored in prior work. This signals a lucrative direction for future research.

ETHICS STATEMENT

The FAIRRET framework was made as a technical tool to help unveil and address a mathematical formalization of fairness in machine learning systems. However, such tools should never be considered a sufficient solution to truly achieve fairness in real-world decision processes because the social, human component of fairness is completely outside the control of this framework (Selbst et al., 2019). There is a significant risk that technologies such as ours may anyway be abused to suggest discriminatory bias has been 'removed' from a decision process without actually addressing underlying injustices (Hoffmann, 2019).

REPRODUCIBILITY

All proofs, i.e. for Table 1, Prop. 1 and Prop. 2, are found in the Appendix A. Appendix B contains additional context for Remarks 1, 2 and 3. Appendix C provides experiments referred to in Sec. 3.2: a visualization of the projections of the projection FAIRRETs and an empirical assessment of their approximation for fewer optimization iterations. It also evaluates the mean SmoothMax loss for smaller batch sizes mentioned in Sec. 3.3. Furthermore, Appendix C extends the main experiment results of Sec. 4.2 by providing the metrics of the other FAIRRETs, the train set results and fairness violations computed for only a single sensitive attribute. Finally, Appendix D further explains the experiment setup already summarized in Sec. 4.1, i.e. the datasets, hyperparameters, the baselines implementation, the computation of the confidence ellipses and runtimes.

The code for our full experiment pipeline is found in the rest of the supplementary material.

The streamlined package is available at `https://github.com/aida-ugent/fairret`. Examples of the code use (at the time of writing the paper) are provided in Appendix E.

ACKNOWLEDGEMENTS

The research leading to these results has received funding from the Special Research Fund (BOF) of Ghent University (BOF20/DOC/144 and BOF20/IBF/117), from the Flemish Government under the "Onderzoeksprogramma Artificiële Intelligentie (AI) Vlaanderen" programme, and from the FWO (project no. G0F9816N, 3G042220, G073924N).

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

# A  PROOFS

## A.1  TABLE 1

Here, we show how the $\alpha_0$, $\alpha_1$, $\beta_0$, and $\beta_1$ functions are derived for each of the fairness notions in Table 1. These fairness notions are typically defined in the partition fairness setting, and so we will make the same assumptions here (see Def. 1). Before discussing each fairness notion separately, we make some general observations.

First, there may be a concern that our fairness constraint, i.e. $\forall k \in [d_s] : \gamma(k; f) = \overline{\gamma}(f)$, requires each group's statistic $\gamma(k; f)$ to equal the overall statistic $\overline{\gamma}(f) \triangleq \gamma(1; f)$, whereas popular surveys of fairness definitions such as by Verma & Rubin (2018) instead only require that each group's statistic is the same, i.e.

$$\forall k, l \in [d_s] : \gamma(k; f) = \gamma(l; f) \tag{A.1}$$

However, these constraints are equivalent.

**Proposition A.1.** *In partition fairness, with $\gamma \in \Gamma$:*

$$\forall k \in [d_s] : \gamma(k; f) = \overline{\gamma}(f) \iff \forall k, l \in [d_s] : \gamma(k; f) = \gamma(l; f) \tag{A.2}$$

*Proof.* The forward ($\implies$) relation is trivial: if all group statistics $\gamma(k; f) = \overline{\gamma}(f)$, then necessarily they are also all equal.

The reverse ($\impliedby$) is less straightforward. Clearly,

$$\forall k, l \in [d_s] : \gamma(k; f) = \gamma(l; f) \iff \exists c \in \mathbb{R} : \forall k \in [d_s] : \gamma(k; f) = c \tag{A.3}$$

It thus suffices to show $c \equiv \overline{\gamma}(f)$ in such cases. Let $g_0(\mathbf{X}, Y) = \alpha_0(\mathbf{X}, Y) + f(\mathbf{X})\beta_0(\mathbf{X}, Y)$ and $g_1(\mathbf{X}, Y) = \alpha_1(\mathbf{X}, Y) + f(\mathbf{X})\beta_1(\mathbf{X}, Y)$. Then

$$\begin{aligned}
\gamma(k; f) &= \frac{\mathbb{E}[S_k g_0(\mathbf{X}, Y)]}{\mathbb{E}[S_k g_1(\mathbf{X}, Y)]} \\
&= \frac{\sum_{X,Y,S} S_k g_0(\mathbf{X}, Y) P(X, Y, S)}{\sum_{X,Y,S} S_k g_1(\mathbf{X}, Y) P(X, Y, S)} \\
&= \frac{\sum_{X,Y} g_0(\mathbf{X}, Y) P(X, Y, S_k = 1)}{\sum_{X,Y} g_1(\mathbf{X}, Y) P(X, Y, S_k = 1)} \\
&= \frac{\sum_{X,Y} g_0(\mathbf{X}, Y) P(X, Y \mid S_k = 1)}{\sum_{X,Y} g_1(\mathbf{X}, Y) P(X, Y \mid S_k = 1)} \\
&= \frac{\mathbb{E}[g_0(\mathbf{X}, Y) \mid S_k = 1]}{\mathbb{E}[g_1(\mathbf{X}, Y) \mid S_k = 1]}
\end{aligned} \tag{A.4}$$

where the third line used the partition fairness assumptions.

Thus, Eq. (A.3) also entails that $\mathbb{E}[g_0(\mathbf{X}, Y) \mid S_k = 1] = c\,\mathbb{E}[g_1(\mathbf{X}, Y) \mid S_k = 1]$.

By the law of total expectation, we now have

$$\begin{aligned}
\overline{\gamma}(f) &= \frac{\mathbb{E}[g_0(\mathbf{X}, Y)]}{\mathbb{E}[g_1(\mathbf{X}, Y)]} \\
&= \frac{\mathbb{E}[\mathbb{E}[g_0(\mathbf{X}, Y) \mid S]]}{\mathbb{E}[\mathbb{E}[g_1(\mathbf{X}, Y) \mid S]]} \\
&= \frac{\sum_k \mathbb{E}[g_0(\mathbf{X}, Y) \mid S_k = 1] P(S_k = 1)}{\sum_k \mathbb{E}[g_1(\mathbf{X}, Y) \mid S_k = 1] P(S_k = 1)} \\
&= \frac{\sum_k c\,\mathbb{E}[g_1(\mathbf{X}, Y) \mid S_k = 1] P(S_k = 1)}{\sum_k \mathbb{E}[g_1(\mathbf{X}, Y) \mid S_k = 1] P(S_k = 1)} \\
&= c
\end{aligned} \tag{A.5}$$

where the third line again used partition fairness. We thus indeed always have $c = \overline{\gamma}(f)$. Applying Eq. (A.3) then completes the proof. $\square$

A second observation used to derive Table 1 is that fairness notions tend to be defined over probabilities involving binary variables. To formulate fairness notions in terms of expectations, we can thus extensively use these binary variables as indicator functions. For example, the frequency of true positives $P(f(\mathbf{X}) = 1, Y = 1)$ is easily counted as $\mathbb{E}[f(\mathbf{X})Y]$.

These observations can be applied to all fairness definitions presented by Verma & Rubin (2018) in their Sections 3.1 and 3.2 and we refer to their work for a detailed interpretation of each definition.

We now show how Table 1 can be constructed by presenting the statistic of each fairness definition and (a simplified form of) their $\gamma$ function under our notation.

- **demographic parity** (a.k.a statistical parity) (Dwork et al., 2012) was already discussed in Example 1. It requires equal positive rates $P(f(\mathbf{X}) = 1 \mid S_k = 1)$, i.e. $\gamma(k; f) = \frac{\mathbb{E}[S_k f(\mathbf{X})]}{\mathbb{E}[S_k]}$.

- **conditional demographic parity** (a.k.a conditional statistical parity) (Corbett-Davies et al., 2017; Wachter et al., 2020) conditions demographic parity on a part of the input. This is typically an input feature with respect to which we are allowed to discriminate. Let $\zeta : \mathbb{R}^{d_x} \to \{0, 1\}$ be an arbitrary function with a binary output. Then the fairness definition requires equal $P(f(\mathbf{X}) = 1 \mid \zeta(\mathbf{X}), S_k = 1)$, i.e. $\gamma(k; f) = \frac{\mathbb{E}[S_k f(\mathbf{X}) \zeta(\mathbf{X})]}{\mathbb{E}[S_k \zeta(\mathbf{X})]}$. Note that this statistic is easily extended to also allow non-binary $\zeta$ functions.

- **equal opportunity** (Hardt et al., 2016) was already discussed in Example 2. It considers positive rates over only the positive samples ($Y = 1$). It thus compares false negative rates, or equivalently true positive rates or recall $P(f(\mathbf{X}) = 1 \mid Y = 1, S_k = 1)$, i.e. $\gamma(k; f) = \frac{\mathbb{E}[S_k f(\mathbf{X}) Y]}{\mathbb{E}[S_k Y]}$.

- **false positive parity** (a.k.a. predictive equality) (Hardt et al., 2016; Corbett-Davies et al., 2017) is similar to equal opportunity, but for the negative class. Hence, it compares true negative rates, or equivalently false positive rates $P(f(\mathbf{X}) = 1 \mid Y = 0, S_k = 1)$, i.e. $\gamma(k; f) = \frac{\mathbb{E}[S_k f(\mathbf{X})(1-Y)]}{\mathbb{E}[S_k(1-Y)]}$. If combined with equal opportunity, it enforces equalized odds (Hardt et al., 2016).

- **predictive parity** (Chouldechova, 2017) was already discussed in Example 3. It compares the positive predictive value, a.k.a the precision $P(Y = 1 \mid f(\mathbf{X}) = 1, S_k = 1)$, i.e. $\gamma(k; f) = \frac{\mathbb{E}[S_k f(\mathbf{X}) Y]}{\mathbb{E}[S_k f(\mathbf{X})]}$.

- **false omission parity** is similar to predictive parity, but for the negative class. It is not explicitly discussed in (Verma & Rubin, 2018), yet it clearly compares false omission rates $P(Y = 1 \mid f(\mathbf{X}) = 0, S_k = 1)$, i.e. $\gamma(k; f) = \frac{\mathbb{E}[S_k(1-f(\mathbf{X}))Y]}{\mathbb{E}[S_k(1-f(\mathbf{X}))]}$.

- **accuracy equality** (Berk et al., 2021) compares accuracies $P(Y = f(\mathbf{X}) \mid S_k = 1)$ across groups. Hence, it computes the relative amount of true positives and true negatives of each group, i.e. $\gamma(k; f) = \frac{\mathbb{E}[S_k(f(\mathbf{X})Y + (1-f(\mathbf{X}))(1-Y))]}{\mathbb{E}[S_k]}$.

- **treatment equality** (Berk et al., 2021) was already discussed in Example 4. It requires the fraction of false negatives over false positives to be equal (or vice versa), and therefore does not represent a probability. Its statistic is thus $\gamma(k; f) = \frac{\mathbb{E}[S_k(1-f(\mathbf{X}))Y]}{\mathbb{E}[S_k f(\mathbf{X})(1-Y)]}$.

## A.2 PROPOSITION 1

*Proof.* Considering the definition of linear-fractional statistics $\gamma \in \Gamma_{\mathrm{LF}}$ in Def. 2, we define the shorthand notations $\alpha(\mathbf{X}, Y, c) = \alpha_0(\mathbf{X}, Y) - c\alpha_1(\mathbf{X}, Y)$ and $\beta(\mathbf{X}, Y, c) = \beta_0(\mathbf{X}, Y) - c\beta_1(\mathbf{X}, Y)$. It is then straightforward to see that

$$
\begin{aligned}
& \gamma(k; f) = c \\
\Longleftrightarrow \quad & \frac{\mathbb{E}[S_k(\alpha_0(\mathbf{X}, Y) + f(\mathbf{X})\beta_0(\mathbf{X}, Y))]}{\mathbb{E}[S_k(\alpha_1(\mathbf{X}, Y) + f(\mathbf{X})\beta_1(\mathbf{X}, Y))]} = c \\
\Longleftrightarrow \quad & \mathbb{E}[S_k(\alpha(\mathbf{X}, Y, c) + f(\mathbf{X})\beta(\mathbf{X}, Y, c))] = 0
\end{aligned}
\tag{A.6}
$$

where the last step uses the linearity of the expectation operator $\mathbb{E}$. The resulting constraint is linear with respect to $f$ since $\alpha$ and $\beta$ are both functions of functions that do not depend on $f$. $\qquad\square$

## A.3 PROPOSITION 2

To show the strictness of the proposed FAIRRETs, we use a separate strategy for violation and projection FAIRRETs.

### A.3.1 VIOLATION FAIRRETS

*Proof.* Per Def. 3, it is easily seen that $\mathbf{v}(h) = \mathbf{0} \iff h \in \mathcal{F}_\gamma$. This also holds in the case where $\overline{\gamma}(h) = 0$, as we can define $\mathbf{v}_k(h) = |\gamma(k; h)|$ there instead. Hence, we only need to show that $R_\gamma(h) = 0 \iff \mathbf{v}(h) = \mathbf{0}$ to show the strictness of violation FAIRRETs $R_\gamma$.

For the Norm FAIRRET, the strictness is obvious: a norm defined over a vector space is always assumed to equal zero only for the vector of zeros, i.e.

$$R_\gamma(h) \triangleq \|\mathbf{v}(h)\| = 0 \iff \mathbf{v}(h) = 0. \tag{A.7}$$

For the SmoothMax FAIRRET, the strictness follows from the $\log d_s$ adjustment term:

$$
\begin{aligned}
R_\gamma(h) &\triangleq \log \sum_{k \in [d_s]} \exp(\mathbf{v}_k(h)) - \log d_s = 0 \\
&\iff \sum_{k \in [d_s]} \exp(\mathbf{v}_k(h)) = d_s \\
&\iff \mathbf{v}(h) = 0
\end{aligned}
\tag{A.8}
$$

where the last step follows from the non-negativity of $v_k$. $\square$

### A.3.2 PROJECTION FAIRRETS

*Proof.* A projection FAIRRET is easily shown to be strict if its divergence measure $D$ satisfies

$$D(f(\mathbf{X}) \parallel h(\mathbf{X})) > 0 \wedge D(f(\mathbf{X}) \parallel h(\mathbf{X})) = 0 \iff f(\mathbf{X}) = h(\mathbf{X}) \tag{A.9}$$

Indeed, if $h \in \mathcal{F}_\gamma$, then also $h \in \mathcal{F}_\gamma(\overline{\gamma}(h))$ by construction. We can then choose $f = h$ and use Eq. (A.9) to get $D(f(\mathbf{X}) \parallel h(\mathbf{X})) = 0$. This shows $h \in \mathcal{F}_\gamma \implies R_\gamma(h) = 0$.

Conversely, the assumed properties of $D$ in Eq. (A.9) entail

$$
\begin{aligned}
R_\gamma(h) &\triangleq \min_{f \in \mathcal{F}_\gamma(\overline{\gamma}(h))} \mathbb{E}[D(f(\mathbf{X}) \parallel h(\mathbf{X}))] = 0 \\
&\iff \exists f \in \mathcal{F}_\gamma(\overline{\gamma}(h)) : \forall \mathbf{X} \in \mathbb{R}^{d_s} : f(\mathbf{X}) = h(\mathbf{X}) \\
&\iff h \in \mathcal{F}_\gamma(\overline{\gamma}(h))
\end{aligned}
\tag{A.10}
$$

where the second line assumes[5] the support of $\mathbf{X}$ equals $\mathbb{R}^{d_s}$. Since $\mathcal{F}_\gamma(\overline{\gamma}(h)) \subset \mathcal{F}_\gamma$, we have $R_\gamma(h) = 0 \implies h \in \mathcal{F}_\gamma$.

To show the strictness of the projection FAIRRETs, it thus suffices to show that the divergences $D$ in Def. 9, 10 and 11 all satisfy the requirements in Eq. (A.9).

It is a well-known property of the Kullback-Leibler divergence $D_{\mathrm{KL}}$ that Eq. (A.9) holds (Csiszar, 1975).

The properties of the $D_{\mathrm{KL}}$-divergence also trivially imply non-negativity for the Jensen-Shannon divergence. Moreover, it entails

$$
\begin{aligned}
D_{\mathrm{JS}}(f(\mathbf{X}) \parallel h(\mathbf{X})) &\triangleq \frac{D_{\mathrm{KL}}(f(\mathbf{X}) \parallel m(\mathbf{X})) + D_{\mathrm{KL}}(h(\mathbf{X}) \parallel m(\mathbf{X}))}{2} = 0 \\
&\iff D_{\mathrm{KL}}(f(\mathbf{X}) \parallel m(\mathbf{X})) = D_{\mathrm{KL}}(h(\mathbf{X}) \parallel m(\mathbf{X})) = 0 \\
&\iff f(\mathbf{X}) = m(\mathbf{X}) = h(\mathbf{X})
\end{aligned}
\tag{A.11}
$$

which means Eq. (A.9) indeed also holds for $D_{\mathrm{JS}}$.

Finally, it is clear that $D_{\mathrm{SED}}$ satisfies Eq. (A.9), as the Euclidean distance is non-negative and only zero for overlapping points. $\square$

---

[5]If the support of $\mathbf{X}$ does not equal $\mathbb{R}^{d_s}$, then we can simply reformulate our framework to only consider function outputs $f(\mathbf{X})$ and $h(\mathbf{X})$ for points $\mathbf{X}$ that *do* lie in its support.

## B    ADDITIONAL DISCUSSION

### B.1    ADDENDUM TO REMARK 1

Observe that, for $\gamma \in \Gamma_{\text{LF}}$:

$$\gamma(k;f) \triangleq \frac{\mathbb{E}[S_k(\alpha_0(\mathbf{X},Y) + f(\mathbf{X})\beta_0(\mathbf{X},Y))]}{\mathbb{E}[S_k(\alpha_1(\mathbf{X},Y) + f(\mathbf{X})\beta_1(\mathbf{X},Y))]} \tag{B.1}$$

and

$$\overline{\gamma}(f) \triangleq \frac{\mathbb{E}[\alpha_0(\mathbf{X},Y) + f(\mathbf{X})\beta_0(\mathbf{X},Y)]}{\mathbb{E}[\alpha_1(\mathbf{X},Y) + f(\mathbf{X})\beta_1(\mathbf{X},Y)]} \tag{B.2}$$

In Remark 1, we (non-exhaustively) mention two cases for $S_k \in \mathbb{R}$ where $\gamma(k;f) = \overline{\gamma}(f)$ holds:

1.  $S_k$ is deterministic, i.e. $S_k \equiv s$ for a constant $s \in \mathbb{R}$. Due to the linearity of the expectation operator, we trivially have $\gamma(k;f) = \overline{\gamma}(f)$. Though this case is degenerate, we should indeed expect a fairness criterion to hold if the random variable $S_k$ expresses no information about an individual (and as such cannot be grounds for discrimination).

2.  $S_k$ has no linear influence on the numerator or denominator of $\gamma$, i.e.

    $$\text{cov}(S_k, \alpha_0(\mathbf{X},Y) + f(\mathbf{X})\beta_0(\mathbf{X},Y)) = \text{cov}(S_k, \alpha_1(\mathbf{X},Y) + f(\mathbf{X})\beta_1(\mathbf{X},Y)) = 0. \tag{B.3}$$

    Indeed, using the definition of the covariance operator, we have

    $$\gamma(k;f) = \frac{\text{cov}(S_k, \alpha_0(\mathbf{X},Y) + f(\mathbf{X})\beta_0(\mathbf{X},Y)) + \mathbb{E}[\alpha_0(\mathbf{X},Y) + f(\mathbf{X})\beta_0(\mathbf{X},Y)]}{\text{cov}(S_k, \alpha_1(\mathbf{X},Y) + f(\mathbf{X})\beta_1(\mathbf{X},Y)) + \mathbb{E}[\alpha_1(\mathbf{X},Y) + f(\mathbf{X})\beta_1(\mathbf{X},Y)]}. \tag{B.4}$$

    Unfortunately, Eq. (B.3) is only a sufficient condition for $\gamma(k;f) = \overline{\gamma}(f)$ in general. Still, it becomes a necessary condition for the DP fairness notion (which always has $\text{cov}(S_k, \alpha_1(\mathbf{X},Y) + f(\mathbf{X})\beta_1(\mathbf{X},Y)) = \text{cov}(S_k, 1) = 0$).

Yet, though we argue $\gamma(k;f) = \overline{\gamma}(f)$ is sensible to enforce for continuous variables $S_k \in \mathbb{R}$, it must be stressed that linear-fractional statistics check *linear* effects of $S$ on $f(\mathbf{X})$. Higher-order moments will thus not be measured.

For example, if $S_0$ denotes 'man-ness' and $S_1$ denotes 'woman-ness', then individuals who identify with a non-binary gender may quantify their sensitive features as $S_0 = 50\%$ and $S_1 = 50\%$. However, linear covariance statistics (like ours) will not consider specific discrimination directed at those with, e.g., $S_1 = 50\%$. Instead, this will be taken into account as 'half as influential' compared to individuals who identify as $S_1 = 100\%$.

### B.2    ADDENDUM TO REMARK 2

In Remark 2, we mention that intersections of demographic groups will not be considered in fairness constraints if partition fairness does not hold.

For example, assume we want to check DP fairness with statistic $\gamma(k;f) = \frac{\mathbb{E}[S_k f(\mathbf{X})]}{\mathbb{E}[S_k]}$ for four uniformly selected samples with scores $(0.7, 0.3, 0.7, 0.3)$ given by $f$, values $(1, 1, 0, 0)$ for sensitive variable $S_0$ and values $(1, 0, 0, 1)$ for sensitive variable $S_1$. Then $\gamma(0;f) = \gamma(1;f) = 0.5$. However, $\frac{\mathbb{E}[S_0 S_1 f(\mathbf{X})]}{\mathbb{E}[S_k]} = 0.7$ and $\frac{\mathbb{E}[(1-S_0)(1-S_1)f(\mathbf{X})]}{\mathbb{E}[S_k]} = 0.3$. Here, the individual with sensitive feature vector $\mathbf{S} = (0,0)$, i.e. at the intersection of $S_0 = 0$ and $S_1 = 0$, thus receives worse scores than the others.

### B.3    ADDENDUM TO REMARK 3

In our discussion of fairness definitions, we assume that we are enforcing constraints on classifiers $f : \mathbb{R}^{d_x} \to \{0, 1\}$, i.e. classifiers with a hard decision boundary. In practice, however, it is common to base decisions off of a parameterized regressor $r : \mathbb{R}^{d_x} \to \mathbb{R}$, e.g. with $r$ a neural network. For example, we can then deterministically collect decisions from $r$ as $f(\mathbf{X}) = \mathbb{1}_{r(\mathbf{X})>0}$ with $\mathbb{1}$ the indicator function. However, such a thresholding function has a gradient of zero with respect to $r$ and a discontinuity in $r(\mathbf{X}) = 0$.

Several works have investigated directly using $r(\mathbf{X})$ in fairness constraints (Zafar et al., 2019). For example in DP fairness, directly computing $\gamma(k; h)$ will already enforce that the mean scores $r(X)$ should be equal across groups. Though this leads to interesting convex optimization problems for a fair $r$, it has been noted that this only observes a relaxed form of actual fairness constraints (Lohaus et al., 2020).

A middle ground is to construct a probabilistic classifier $f$ as a Bernoulli distribution with parameter $h(\mathbf{X}) = \sigma(r(\mathbf{X}))$ where $\sigma$ denotes the logistic function $\sigma(s) = (1 + \exp(-s))^{-1}$. Hence, $f(\mathbf{X})$ is fully specified through $P(f(\mathbf{X}) = 1 \mid \mathbf{X}) = h(\mathbf{X})$. In the definition of $\gamma$, we can then simply replace the classification function $f$ in the expectation by the scoring model $h$, because $\mathbb{E}[\mathbb{E}[f(\mathbf{X}) \mid \mathbf{X}]] = \mathbb{E}[h(\mathbf{X})]$. Still, the fact that a probabilistic classifier is now not only dependent on $\mathbf{X}$ but also on the randomness involved in sampling $f(\mathbf{X})$ introduces some noise into the decision process that we may wish to avoid. There is also the danger that in practice, real-world decisions would still be made according to a hard threshold on the scores $h(\mathbf{X})$, e.g. on $f(\mathbf{X}) = \mathbb{1}_{h(\mathbf{X})>0.5}$.

If this becomes an issue, then we could make use of recent work (Padh et al., 2021; Bendekgey & Sudderth) that has investigated surrogate functions that are better approximations of the thresholding function $\mathbb{1}_{\cdot>0}$ than the logistic function $\sigma$. For example, a suitable surrogate is the scaled logistic function $\sigma_a(s) \triangleq \sigma(as)$, since $\lim_{a\to\infty} \sigma_a(s) = \mathbb{1}_{s>0}$ for $s \neq 0$. However, note also that $\lim_{a\to\infty} \frac{\partial}{\partial s}\sigma_a(s) = 0$. Surrogate functions like $\sigma_a$ thus allow us to trade-off the hardness of the classification with the quality of its gradient.

## C ADDITIONAL RESULTS

### C.1 APPROXIMATE PROJECTION

In Sec. 3.2, we suggest that the convex optimization of the projections $f^*$ can already converge quite well after only 10 iterations. By placing such a limit, we can significantly reduce the computational cost of $R_\gamma(h)$, though $R_\gamma(h)$ will be an overestimate (as projections $f^*$ with a smaller divergence to $h$ may not have been found yet).

In Fig. C.1, we observe that training with at most 10 iterations per projection is indeed enough to minimize the DP violation while also being much faster to compute. Unexpectedly, we even observe the DP violation to be slightly lower after training with at most 10 iterations than with more than 10, which could indicate that fewer iterations can have a positive effect on the training process.

### C.2 PROJECTION VISUALIZATION

In Fig. C.2, we visualize the projected distributions for the example in Fig. 1. The $D_{\mathrm{KL}}$- and $D_{\mathrm{JS}}$-projections appear to transform the shape of the probability distribution, whereas the $D_{\mathrm{SED}}$-projection appear to linearly shift the probabilities within a group.

For the latter, this appears to form a problem, because its behavior leads to a large gap in densities between $h$ and the projection $f^*$ at the edges of the $[0, 1]$ interval. For example for $S_0 = 1$, the scores are linearly shifted to the left (lower probability scores), meaning that no scores are left in the high probability range, and too many are allocated to the low probability range. The 'blocking' of this shift on the low end for $S_0 = 1$ and on the high end for $S_1 = 1$ is in fact the reason for the 'crack' in the gradients of the $D_{\mathrm{SED}}$-projection in Fig. 1, since the probabilities of $h$ cannot be projected beyond these edges.

### C.3 MINI-BATCHING THE FAIRRET

In Sec. 3.3, we propose that FAIRRETs $R_\gamma$ can be minimized using the same mini-batches that we use to minimize the cross-entropy loss $\mathcal{L}_Y$. However, batches clearly need to be large enough to adequately represent the imbalances for all sensitive features $\mathbf{S}$. Hence, we report an experiment in Fig. C.3 where we take an unfair classifier $h$ and compute the mean SmoothMax loss over batch sizes with increasing granularity.

From a batch size of approximately 1024, the mean loss over all batches already closely matches the SmoothMax loss computed over the full test set (39 133 samples). Note that for very small batch

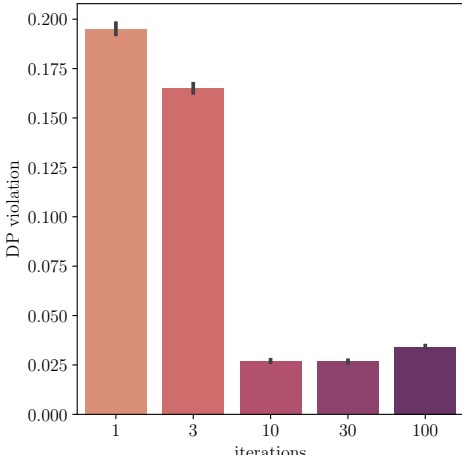 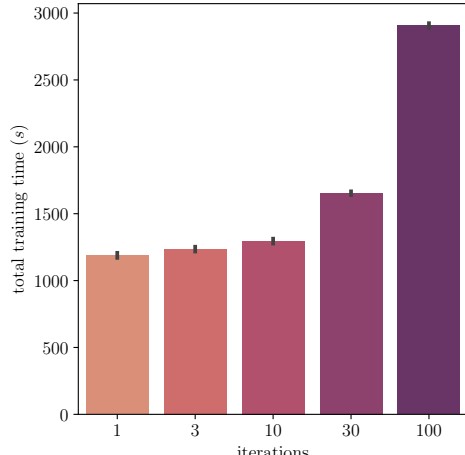

Figure C.1: (**left**) Test set DP violation, with a similar experiment setup as Fig. 2 on the ACSIncome dataset. Each result bar results from a separate training run with the $D_{KL}$-projection fairret that was minimizing the DP violation. The configurations only differ in the maximum number of iterations used in the convex optimizations that compute the actual $D_{KL}$-projections $f^*$ (see Sec. 3.2). (**right**) The total training time of these runs with standard error.

sizes, the loss becomes mostly meaningless, as batches will not even contain all members of each protected group. To be on the safe side, we use a batch size of 4096 in all our experiments.

## C.4 OTHER FAIRRET RESULTS

The test set results on the Norm, $D_{JS}$-projection and $D_{SED}$-projection FAIRRETs are shown in Fig. C.4. The Norm FAIRRET follows a very similar gradient as the SmoothMax FAIRRET, whereas the $D_{JS}$-projection and $D_{SED}$-projection give similar results as the $D_{KL}$-projection.

## C.5 TRAIN SET RESULTS

In Fig. C.5, we show the train set results for our main experiments. These follow the same trends as the test set results, though with higher AUROC scores and lower fairness violations. The most important difference is the performance of the SmoothMax fairret, e.g. on the CreditCard dataset, which obtains a significantly lower fairness violation for the difficult PP and TE fairness notions than it did on the test set.

## C.6 FAIRNESS VIOLATIONS FOR A SINGLE SENSITIVE ATTRIBUTE

The FFB implementation (and indeed most fairness tools) only mitigate bias with respect to a single, categorical sensitive feature. In our datasets (see Sec. D.1), these were the following:

- Bank: marital status
- CreditCard: sex
- LawSchool: sex
- ACSIncome: sex

The fairness violations for these single features are shown in Fig. C.6. Importantly, the FAIRRET experiments were redone by training with only this single feature in mind when computing the FAIRRET loss. Even for the DP fairness notion, the FAIRRETs remain competitive and the $D_{KL}$-projection obtains the best trade-off.

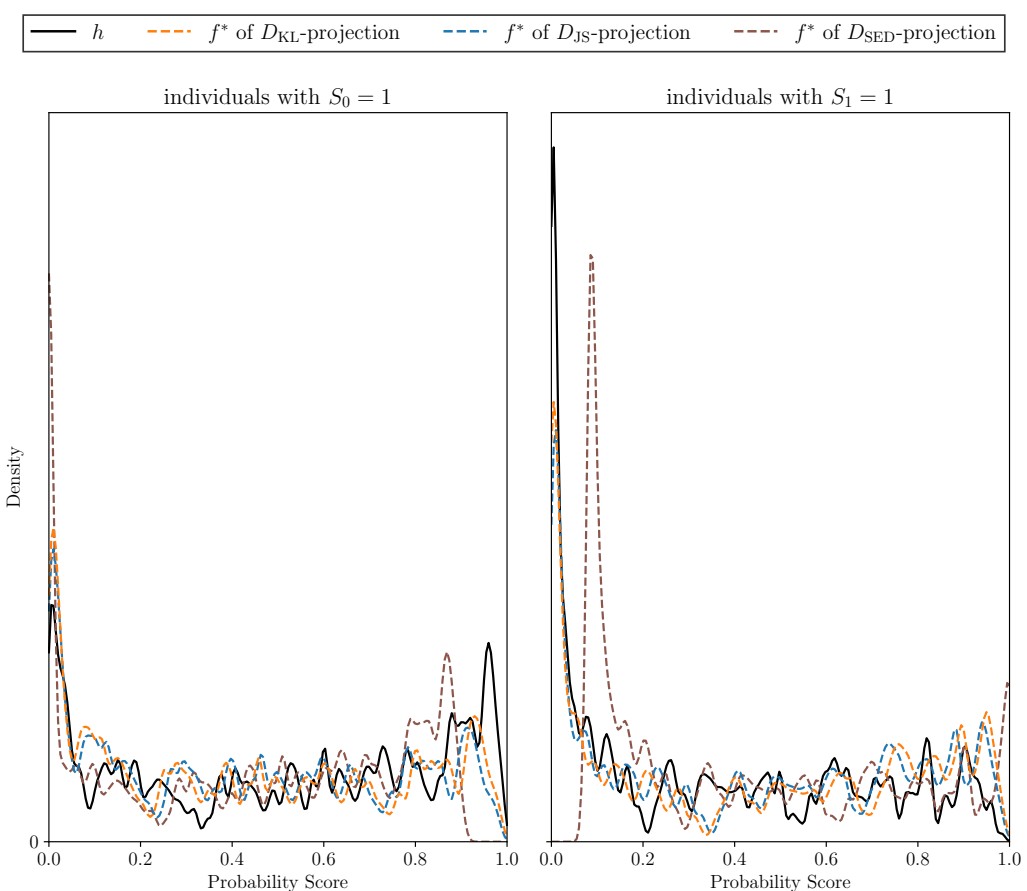

Figure C.2: Starting from the same setup as in Fig. 1, we show the probability scores of both $h$ (full line) and the projected distributions $f^*$ of each projection FAIRRET (dotted lines). The $y$-axis shows the KDE densities of these scores, all on the same scale.

# D    ADDITIONAL EXPERIMENT DETAILS

## D.1    DATASETS

We used four different datasets for the evaluation of our framework, namely the Bank Marketing dataset (Moro et al., 2014), Credit card clients dataset (Yeh & hui Lien, 2009), Law School Admissions dataset [6] and the ACSIncome dataset from Folktables (Ding et al., 2021). Their main advantages are their range of sensitive attributes, their recency and their curation quality.

We deviate from the normal practice of using the German Credit and Adult data sets as advised by Fabris et al. (Fabris et al., 2022) due to the "contrived prediction tasks, noisy data, severe coding mistakes, limitations in encoding sensitive attributes, and age" of these datasets.

### D.1.1    BANK

The *Bank marketing* dataset was collected by a Portuguese bank between May 2008 and June 2013. The dataset includes all information the bank has on a client, information about the previous attempt for the client to subscribe for a long-term deposit, some economic information and if the client decided to go for a long-term deposit following the telephone call.

---

[6]Curated and published by the SEAPHE project

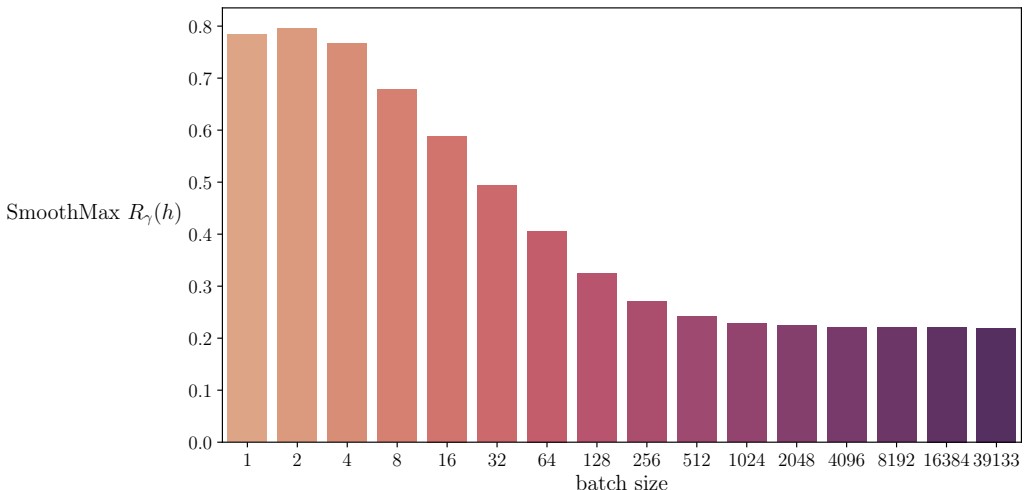

Figure C.3: Test set SmoothMax loss $R_\gamma(h)$ with $\gamma$ the positive rate statistic (enforcing the DP notion), computed for an unfair model $h$ trained with the same setup as in Fig. 1. Each loss is computed over the entire test dataset, but chunked using different batch sizes. For smaller batch sizes, the mean SmoothMax loss is an overestimate of the actual SmoothMax loss computed over all 39 133 samples.

The dataset itself contains the information of 41 108 telephone contacts. Important to note for this dataset is that the outcomes are severely unbalanced with only 4 640 of outcomes belonging to the positive class.

The sensitive attributes which were used as such in this dataset were the age and marital status of a person. A person's eduction was not included as a sensitive attribute as discriminating on that value could arguably be justifiable in this situation.

During preprocessing, we dropped five features: three relating to the outcome of the previous marketing campaign as often there was no record of a previous contact and two features relating to the date when the current marketing call took place. If the value of certain features were unknown then they were were mapped onto the 'False' value, except for the case if the marital status of the person was unknown. In that instance the row was simply dropped, since this only occurred for 80 samples in the entire dataset.

### D.1.2 CREDITCARD

The *Credit Card clients* dataset is data from a bank in Taiwan in October 2005. The goal is to predict whether a client would default on their credit in the next month. The features include the allocated credit of the client, personal information, the status of previous payments, amounts of previous payments and the amounts on previous bill statements.

In total the dataset contains 30 000 records. In 5529 of these instances the client defaulted in the next month.

This dataset contains a wide range of sensitive attributes, namely sex, education, marital status and age.

A total of 522 samples were removed from the dataset as they contained values unspecified in the documentation of the data or if their eduction status fell under the category others. That was mainly done as only 123 samples were of this category, making it too small to maintain adequate statistical power for the fairness measure of the group.

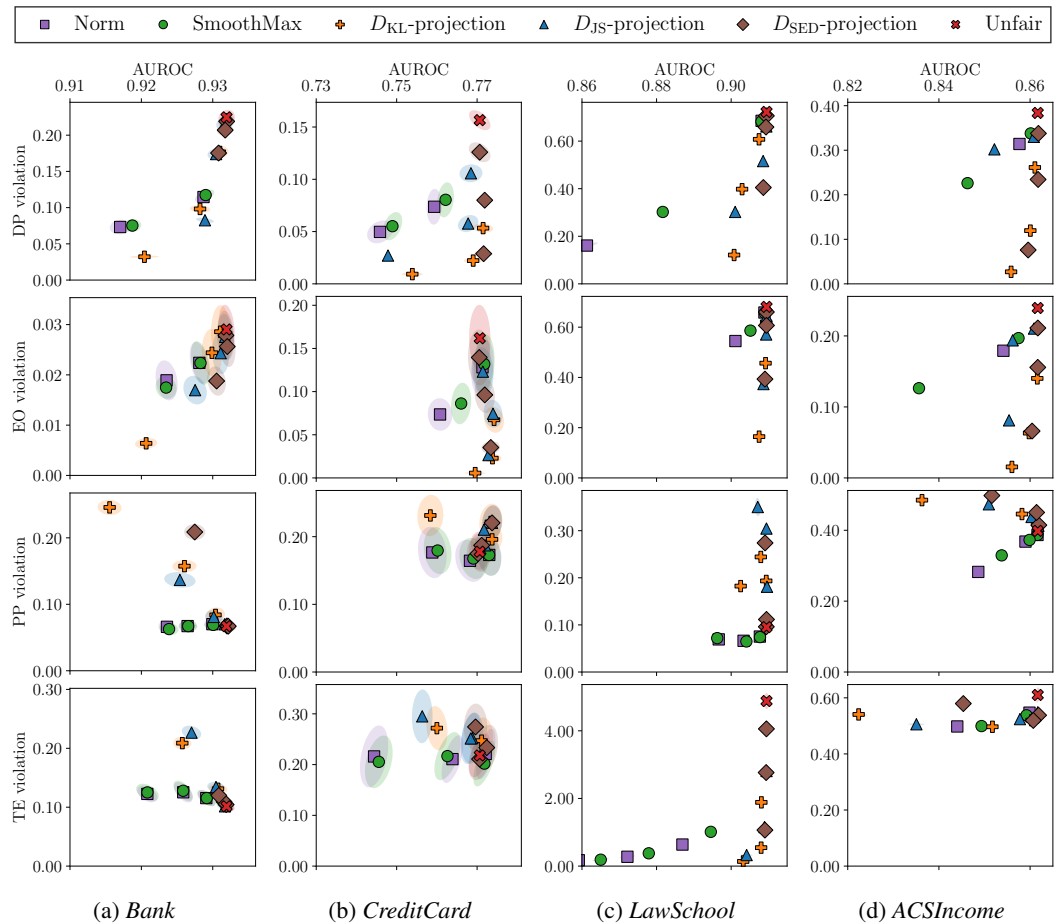

Figure C.4: Test set results for the experiments in Fig. 2, but with different FAIRRETs.

### D.1.3 LAWSCHOOL

The *Law School admissions* dataset contains information about whether a student was accepted to the law school, which we tried to predict in our experiments. The dataset also contains the student's race, gender, whether they live where they applied to law school, what college they applied to, the year they did this, and their LSAT and GPA scores.

Although this dataset is hand-curated by the SEAPHE project, it is still fairly sizeable with 65 535 samples. Only 24.84% of these samples had a positive outcome.

The dataset only contains two sensitive attributes: race and sex. This is the only dataset where the age features are not available. Interestingly, the range of values for the race attribute is fairly balanced across all races, except for white which is overly represented with 39 742 samples.

In the preprocessing step only the year of the application column was not used as a feature.

### D.1.4 ACSINCOME

The *ACSIncome* dataset has the familiar goal of predicting whether an individual's annual income is above $50 000. The source of the data is the US census study. This only includes individuals above the age of 16, who indicate having worked at least one hour per week and reported an income above $100. A myriad of personal information is available in the dataset.

The dataset is significantly larger than the others used, with 195 665 samples. It is also more balanced as it has 80 335 positive samples.

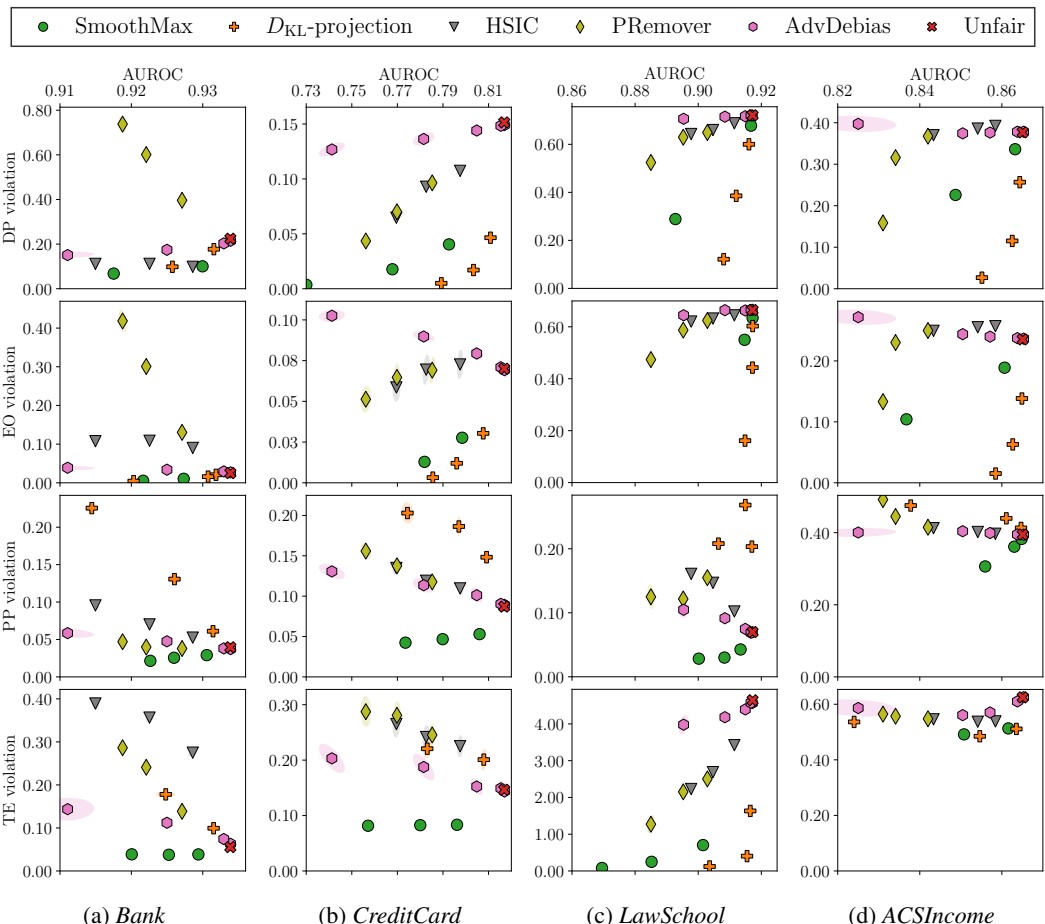

Figure C.5: The *train* set results corresponding with Fig. 2.

Four sensitive attributes are used for the calculations. In this case age, marital status, sex and race. Information about employment and education is not included as sensitive attributes.

Due to the large amount of race groups in the survey, it is necessary to simplify this trait in order to have each group be at least a total of 1% of the data set to guarantee statistical significance for each group.

## D.2  HYPERPARAMETERS

In addition to the hyperparameters discussed in Sec. 4.1, we also mention that our model was optimized with the *Adam* optimizer implementation of PyTorch, with a learning rate of 0.001 and a batch size of 4096. The loss was minimized over 100 epochs, with $\lambda = 0$ for the first 20 to avoid constraining $h$ before it learns anything.

To find these hyperparameters, we took the 80%/20% train/test split already generated for each seed, and further divided the train set into a smaller train set and a validation set with relative sizes 80% and 20% respectively. Keeping the FAIRRET strength $\lambda = 0$, we performed a grid search on the neural architecture in range

$$[[], [128], [128, 32], [128, 64, 32], [256, 128, 32], [512, 256, 128, 32], [512, 512, 256, 128, 32]] \tag{D.1}$$

in combination with the learning rate in range $[0.01, 0.005, 0.001, 0.0005, 0.0001]$. We then selected the combination with the best AUROC on the validation set of the Bank dataset. The number of epochs and the batch size were tuned manually based on the convergence properties of the validation AUROC.

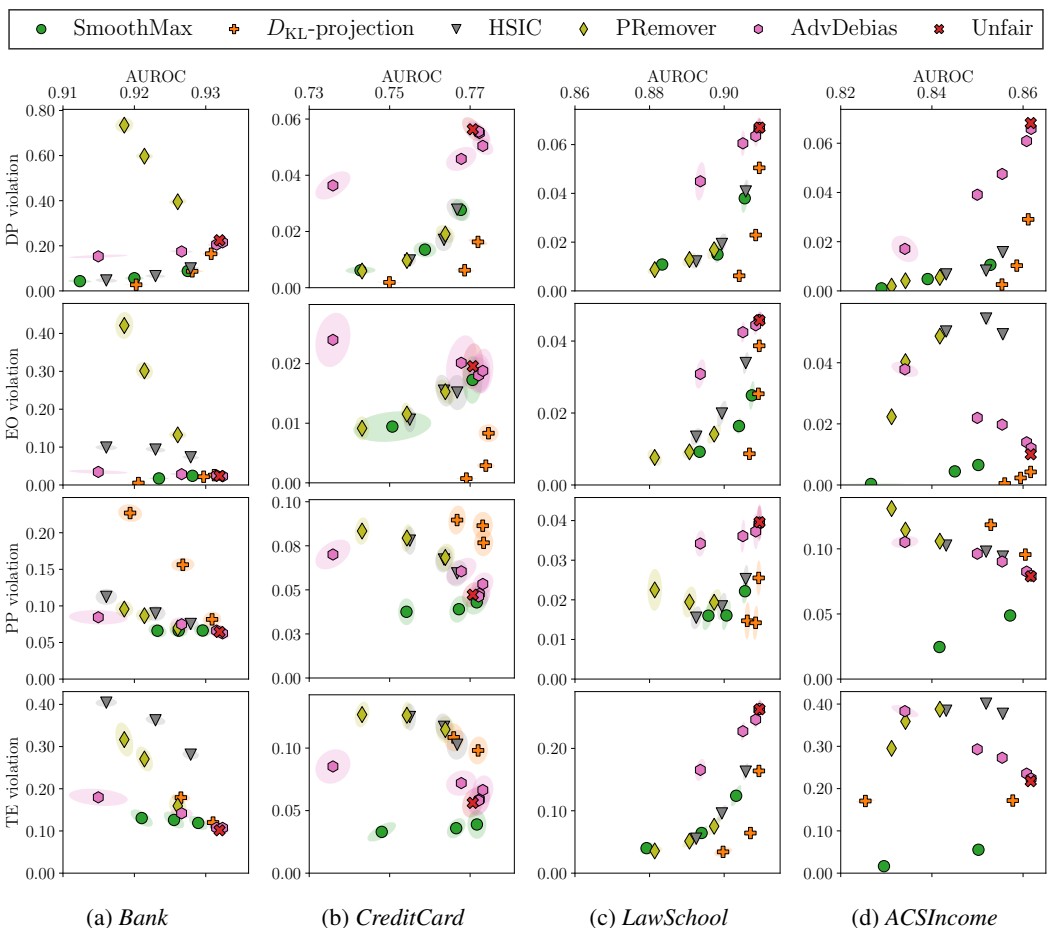

Figure C.6: Test set results for the experiments in Fig. 2, but with fairness violations computed for a single sensitive feature. For these results, FAIRRET experiments were redone and optimized for only this feature specifically.

## D.3 FFB BASELINES

For all FFB (Han et al., 2023) baselines, we used the publicly available implementation[7] with only minor adjustments to make them fit in our experiment pipeline.

From their implementation, we used the following baselines.

- HSIC minimizes the Hilbert-Schmidt Independence Criterion between the model's prediction probabilities and the sensitive attributes (Pérez-Suay et al., 2017).

- PRemover minimizes the mutual information between the model's prediction probabilities and the sensitive attributes (Kamishima et al., 2012).

- AdvDebias maximizes an adversary's cross entropy between its predictions for the sensitive attribute and the actual sensitive attribute, given the last hidden layer of $h$ Adel et al. (2019). after trying several configurations, we achieved the most stable results with an adversarial net of one hidden layer of size 32.

We again stress these implementations only optimize for the specific fairness notion listed above and only for a single categorical sensitive attribute.

---

[7]https://github.com/ahxt/fair_fairness_benchmark/commit/
abec4de80455831ce8d2e158629dfb738a572201

At the time of writing, the FFB also contains implementations for a regularization term that minimizes the gap to obtain DP and EO, but this is highly similar to our violation fairrets and would not be an informative comparison.

## D.4 Confidence Ellipses

The confidence ellipses we use in Fig. 1, Fig. C.4 and Fig. C.5 are uncommon in machine learning literature. Yet, they work well for our purpose of comparing trade-offs between metrics that may be noisy depending on randomness during training and dataset split selection.

Recall that 1-dimensional confidence intervals typically assume a mean estimator to be normally distributed. The confidence interval then denotes the uncertainty of the sample mean using the standard error. Similarly, confidence ellipses assume a 2-dimensional point, i.e. the 2-dimensional mean estimator, to have a multivariate normal distribution that can be characterized through the sample mean and standard error statistics.

Our implementation of the confidence ellipses follows a featured implementation on `matplotlib`[8]. However, a crucial difference is that this implementation computes a confidence interval for a 2-dimensional random variable based on the covariance matrix for the standard *deviation* of samples of that variable. Following observations by Schubert and Kirchner (Schubert & Kirchner, 2014), we instead want to show the uncertainty of the mean estimator, which should use the standard deviation of that estimator, i.e. the covariance for the standard *error*. This is accomplished by dividing the covariance matrix in the `matplotlib` implementation by the number of seeds (5) we use in our experiments.

## D.5 Computation Cost and Runtimes

We report some of the runtimes in Fig. D.1 that illustrate the difference in computational cost between FAIRRETs and baselines during the main experiments of Sec. 4. Note that both our FAIRRET implementation and the FFB implementation were designed for intuitive use in research and not yet optimized for runtime speed.

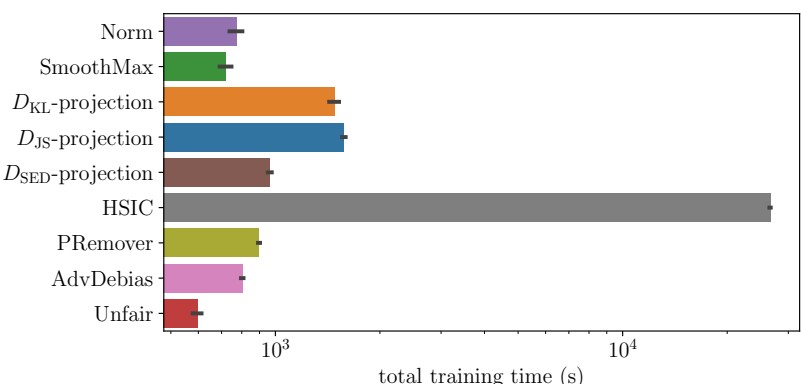

Figure D.1: Runtimes for the ACSIncome dataset experiments discussed in Sec. 4 with strength $\lambda = 1$ (except for the *Unfair* baseline). The FAIRRETS were optimizing the DP fairness notion. Note the log scale.

All experiments in Sec. 4 were conducted on an internal server equipped with a 12 Core Intel(R) Xeon(R) Gold processor and 256 GB of RAM. All experiments, including preliminary and failed experiments, cost approximately 100 hours per CPU.

```python
import torch
import torch.nn.functional as F

from fairret.statistic import TruePositiveRate
from fairret.loss.violation import NormLoss

# The TruePositiveRate class is a subclass of LinearFractionalStatistic.
statistic = TruePositiveRate()

# The fairret modules accept any LinearFractionalStatistic instance.
fairret = NormLoss(statistic)
fairret_strength = 1.0

def train_epoch(train_loader, model, optimizer):
    for feat, sens, target in train_loader:
        optimizer.zero_grad()

        logit = model(feat)
        bce_loss = F.binary_cross_entropy_with_logits(logit, target)
        fairret_loss = fairret(logit, feat, sens, target)
        loss = bce_loss + fairret_strength * fairret_loss
        loss.backward()

        optimizer.step()
```

Listing E.1: Example use of the FAIRRET library in a simple PyTorch setup.

## E  CODE USE EXAMPLES

Listing E.1 displays a code example of how the FAIRRET can easily be deployed in a typical PyTorch (Paszke et al., 2019) setup.  It suffices to simply load a subclass of LinearFractionalStatistic and pass it on to a FAIRRET implementation instance such as NormLoss (as defined in Def. 7). The FAIRRET is then used to compute the quantification of unfairness as a loss like any other in PyTorch. In this case, we use the true positive rate statistic to pursue the fairness notion of equalized opportunity (EO).

In Listing E.2, we provide an example implementation of a LinearFractionalStatistic class, which only entails the specification of the $\alpha$ and $\beta$ functions as in Table 1.

---

[8]https://matplotlib.org/3.7.0/gallery/statistics/confidence_ellipse.html.

```python
import torch

from fairret.statistic import LinearFractionalStatistic

class Accuracy(LinearFractionalStatistic, name="acc"):
    # alpha_0
    def nom_intercept(self, feat, label):
        return 1 - label

    # beta_0
    def nom_slope(self, feat, label):
        return 2 * label - 1

    # alpha_1
    def denom_intercept(self, feat, label):
        return torch.ones(feat.shape[0])

    # beta_1
    def denom_slope(self, feat, label):
        return torch.zeros(feat.shape[0])
```

Listing E.2: Example implementation of a custom LinearFractionalStatistic.

