# OpenReview forum: "fairret: a Framework for Differentiable Fairness Regularization Terms"
_ICLR.cc/2024/Conference — ICLR 2024 poster_

### Official Review · Reviewer_BABL · 2023-10-12

**Soundness:** 1 poor
**Presentation:** 2 fair
**Contribution:** 1 poor
**Rating:** 3
**Confidence:** 5

**Summary:**

Paper claims that current tools for machine learning fairness only admit a limited range of fairness definitions and have seen little integration with automatic differentiation libraries.
For this reason they introduce a framework of fairness regularization terms which quantify bias as modular objectives that are easily integrated in automatic differentiation pipelines.

**Strengths:**

The overview, FAIRRET, and results are interesting and valuable.

**Weaknesses:**

Authors claim is an overstatement in fact there are plenty of work that proposes differentiable (and in some case even convex) regularisers.
State of the art is largely incomplete.

**Questions:**

There are plenty of work that proposes differentiable (and in some case even convex) regularisers (e.g. [1] but in ICML, NeurIPS, etc. you can find plenty of work on this). Paper should elaborate on that.

[1] Exploiting MMD and Sinkhorn Divergences for Fair and Transferable Representation Learning, NeurIPS 2020.
[2] Deep Fair Models for Complex Data: Graphs Labeling and Explainable Face Recognition, Neurocomputing 2021.

---

> ### Author Response · Authors · 2023-11-15
>
> We thank the Reviewer for sharing their thoughts.
>
> The area of *fair representation learning* is indeed a line of work that was not acknowledged in our Related Work section. Note that we *do* compare against this class of methods by having an adversarial method (Adel et al. 2019) as a baseline, which also tries to find fair representations in a hidden layer. However, a broader overview of fair representation learning was indeed warranted, and so we added additional references in our Related Work.
>
> Yet, this line of research does not refute our claim that “current tools for machine learning fairness (1) only admit a limited range of fairness definitions and (2) have seen little integration with automatic differentiation libraries”. Though methods for fair representation learning typically use automatic differentiation (2), they cannot directly pursue a wide range of fairness definitions (1). For example, both your provided references (Oneto et al. 2020; Franco et al. 2022) only address the fairness notion of demographic parity. The reason is clear: fair representation learning focuses on the distribution of dense, internal representations. However, fairness definitions are typically model-agnostic and only consider the output of the model. It is these latter definitions that we generally pursue with differentiable regularizers in our framework, which is also model-agnostic (requiring only that the model is a differentiable, probabilistic classifier).
>
> Moreover, the claim “current tools for machine learning fairness have seen little integration with automatic differentiation libraries” clearly holds for most fairness toolkits, such as Fairlearn and AIF360. These require models to follow the interface of `scikit-learn` Estimators. Instead, fairrets are modular objectives that can be plugged into a training step. We show an example of this in the revised Appendix E.
>
> Besides the questioning of this single claim regarding the state-of-the-art, the Reviewer did not provide further arguments to support their low rating.
>
> **References**
>
> Adel, T., Valera, I., Ghahramani, Z., & Weller, A. (2019, July). One-network adversarial fairness. In Proceedings of the AAAI Conference on Artificial Intelligence (Vol. 33, No. 01, pp. 2412-2420).
>
> Oneto, L., Donini, M., Luise, G., Ciliberto, C., Maurer, A., & Pontil, M. (2020). Exploiting mmd and sinkhorn divergences for fair and transferable representation learning. Advances in Neural Information Processing Systems, 33, 15360-15370.
>
> Franco, D., Navarin, N., Donini, M., Anguita, D., & Oneto, L. (2022). Deep fair models for complex data: Graphs labeling and explainable face recognition. Neurocomputing, 470, 318-334.

---

> > ### Comment · Reviewer_BABL · 2023-11-16
> >
> > Dear authors, thanks for the reply.
> > Regarding comparison with baselines i think that  (Adel et al. 2019) is not enough nether a state-of-the-art baseline given the large amount of more recent works.
> > Regarding the comment on the fact that just demographic parity is tested, i think is not accurate. Most group fairness definitions (e.g., equal odds, equal opportunity, uncorellation) can be defined under the same hat bus simply constraining the distribution of the representation to be close for a subset of the data distribution (e.g., for equal opportunity the distribution of the representation of the male labeled with +1 and the distribution of the representation of the female labeled with +1).
> > Regarding the claim of differentiability, there is a number of works which propose both differentiable or even convex relaxation of the fairness definitions with theoretical properties so the statement for me is still too much.
> > All these comments led me to my low rank: paper novelty, in my opinion, is limited since large amount of related works are not properly compared to the proposal neither theoretically nor empirically.

---

> > > ### Author Response · Authors · 2023-11-16
> > >
> > > We thank the Reviewer for their fast reply and we highly appreciate that they elaborated on their concerns. It appears there is a misunderstanding on where the intended novelty claims of our paper lie. This is likely due to our ambiguous use of the term “fairness tools” in the abstract (which was already remarked upon by Reviewer hHUL). **Please allow us to clear up our novelty claim now - in the rebuttal and in the revised abstract and introduction.**
> > >
> > > Our claim is *not* that fairness research has seen little integration with automatic differentiation, nor do we claim there are few differentiable relaxations of fairness definitions. Rather, fairness toolkits that practitioners and researchers can easily deploy in existing pipelines (see AIF360 and Fairlearn) have thus far seen little interaction with this wealth of research into differentiable fairness methods. This is surprising, given that these methods naturally lend themselves to modular, flexible code. Nevertheless, little such code exists. The current focus in the field seems to lie with improving trade-offs and providing theoretical guarantees, rather than putting a similar amount of resources into enabling the practical use.
> > >
> > > Our goal is therefore to consolidate, simplify, and generalize the wealth of research into fairness regularization terms such the innovations of this field *can* be put into practice more easily. Hence, **the most relevant SOTA for our paper is not the most advanced differentiable fairness method, but rather the most practical implementation of differentiable fairness methods**. The most related is thus the recently proposed FFB library, which we compare with extensively.
> > >
> > > Please note that we have modified the discussed claim in the revised paper.

---

> ### Comment · Area_Chair_mCfz · 2023-12-05
>
> Dear Reviewer BABL,
>
> The authors have answered your comments. Please read their responses as soon as possible (if you haven't) and reply here if you think the authors have addressed your concerns (maybe some of them). Please update your rating if you have changed your evaluation, or at least mention that you want to keep your rating after reading.
>
> Thank you so much.
>
> Area Chair

---

### Official Review · Reviewer_hHUL · 2023-10-23

**Soundness:** 3 good
**Presentation:** 3 good
**Contribution:** 2 fair
**Rating:** 8
**Confidence:** 4

**Summary:**

In this paper, the authors study the problem of fairness in ML. To be specific, they introduce a formal framework which facilitates defining regularization terms for fairness which can be minimized using auto-differentiation tools. The authors report significant improvements over baselines.

**Strengths:**

+ The paper provides a good formal coverage of the fundamental concepts.

**Weaknesses:**

1. I have concerns about the novelty. It appears that the paper's novelty is a formal fairness regularization framework. The paper criticizes existing frameworks for not being formal and limited in terms of fairness definitions. However, the paper does not showcase what the benefit of this formal framework is. Moreover, it is not clear why FFB or FairTorch cannot be extended to include more fairness definitions.

2. I find "fairness tools" misleading. This is sometimes used to refer to fairness measures,  sometimes to their approximations as regularization terms and sometimes to their implementations.


Minor comments:
- I would recommend following the following guide while writing equations:
http://www.ai.mit.edu/courses/6.899/papers/mermin.pdf

**Questions:**

Please see Weaknesses.

**After Rebuttal**

I've read the comments provided by other reviewers and the responses by the authors. I find that the authors have sufficiently addressed my concerns. Looking at the sample code in Appendix E and the implementations in FFB, I see the contribution of the paper better. I think it will be beneficial for the community. Therefore, I've changed my recommendation.

---

> ### Author Response · Authors · 2023-11-15
>
> We would like to thank the Reviewer for their review. Though critical, we appreciate the Reviewer directly engages with the intended goal and significance of our paper.
>
> **Weaknesses**
>
> 1. *On the novelty/significance*
>
> We admit that the benefit of our formal framework was not explicitly stated in the paper; please allow us to do so now - in this rebuttal as well as in the revised version of the paper. Prior work such as by Padh et al. (2021) or Wick and Tristan (2019) on fairness regularizers only admits a few select fairness definitions and for a single, categorical sensitive feature. By formally considering widely applicable fairness definitions and focussing on regularizers that can obtain those general definitions, we can both consolidate existing work and serve as a foundation for future results that can benefit from this generality. Future regularizers need only show they strictly quantify the discrepancy between our statistics to enjoy the same wide applicability.
>
> This is already evident in our empirical results: our fairret implementations significantly outperform existing implementations due to their modularity in how fairness is defined. The mathematical simplicity of our interface also translates to easily extensible code, as shown in the newly added Appendix E that contains a code snippet of how fairrets can easily be used in automatic differentiation and how novel fairness (linear-fractional) statistics can be defined. In addition to Appendix E, we now also more clearly state the intended benefit of our formal framework in Section 1 of the revised main paper.
>
> *On the differences with FFB and FairTorch*
>
> The structure of both FFB and FairTorch does not lend itself to the general applicability of the fairret framework. Both these packages have their fairness regularization terms hardcoded with a select few fairness measures. In our fairret implementation, the regularization term is completely modular with respect to the fairness measure. A user can thus easily create their own fairness measure and immediately use them in the framework. Again, an example of this use can now be found in Appendix E. Hence, our framework is a generalization of the methods implemented in FFB and FairTorch. Please note we reached out to (but failed to receive a response from) one of the authors of FFB and will continue to seek collaboration, as their extensive experiments are complementary (rather than overlapping) with our work.
>
>
> 2. *The use of fairness tools*
>
> We appreciate the Reviewer’s comments and streamlined the use of the term ‘fairness tools’ in order to mitigate any confusion.
>
> *Minor comments*
>
> We thank the Reviewer for this valuable resource. We’ve adjusted the use of equations in the paper according to the article’s suggestions.
>
> **References**
>
> Padh, K., Antognini, D., Lejal-Glaude, E., Faltings, B., & Musat, C. (2021, December). Addressing fairness in classification with a model-agnostic multi-objective algorithm. In Uncertainty in artificial intelligence (pp. 600-609). PMLR.
>
> Wick, M., & Tristan, J. B. (2019). Unlocking fairness: a trade-off revisited. Advances in neural information processing systems, 32.

---

> > ### Comment · Reviewer_hHUL · 2023-11-16
> > **Re: Official Comment by Authors**
> >
> > Dear authors,
> >
> > Thank you for the detailed and rich response. Based on your response, I looked at your example code in Appendix E and the implementations in FFB. I concur that your framework makes it easier for other researchers to integrate new measures of fairness and regularize their networks accordingly. I find this approach positive and useful for the community.
> >
> > I will increase my original rating.
> >
> > Good luck

---

> > > ### Author Response · Authors · 2023-11-16
> > >
> > > Dear Reviewer hHUL,
> > >
> > > Thank you for taking the time to make this additional comparison. We are very glad to learn it has led to a positive evaluation and will keep your feedback in mind moving forward.

---

### Official Review · Reviewer_eBDd · 2023-10-30

**Soundness:** 3 good
**Presentation:** 3 good
**Contribution:** 2 fair
**Rating:** 5
**Confidence:** 4

**Summary:**

The authors present a general framework for formulating fairness regularization terms which are differentiable. Their framework encompasses a wide range of group fairness notions. The authors then propose a series of regularizers for enforcing the fairness definitions. Finally, an experimental evaluation is given.

**Strengths:**

I think the authors propose a valuable contribution to the fairness community. Specifically, I appreciate the effort the authors make on combining multiple fairness notions into a general framework. I also think that providing a python package can be valuable for research and adaptation of fairness in ML.

**Weaknesses:**

The theoretical contribution of this work is in my opinion quite limited. While combining different fairness notions is valuable, I do not think that this is in itself a theoretical contribution. For instance, if in (3) you fix $\overline{\gamma}(h)=c$, then a norm based regularizer would be convex in $f(X)$. Thus if the model is linear, you would get a convex regularizer. Combine this with a convex $\mathcal{L}_Y$ and your problem is convex in the model parameters. This would allow you to get fast convergence rates.

**Questions:**

Some minor questions:
- Why do you call your method "Partition Fairness" and not "Group Fairness" as is common in the literature?
- Regarding continuous sensitive variables. It seems that this approach only captures linear correlation between the sensitive variables and the score. Is this correct?

---

> ### Author Response · Authors · 2023-11-15
>
> We thank the Reviewer for the time they invested into reviewing the paper and for providing insightful comments.
>
> **Weaknesses**
>
> *Novelty of theoretical contribution*
>
> We agree that the significance of our paper is not found in theoretical results. Rather, we provide an algorithmic contribution by combining recent advances into a general, formal framework, for which we provide a thorough empirical analysis. The flexibility of this framework can have it serve as a foundation for future work on fairness regularization. This is clarified in the revised Section 1 of the paper. In the newly added Appendix E, we demonstrate that fairrets are easy to use in typical PyTorch settings and allow an easy extension to new linear-fractional fairness statistics.
>
> *Linear model leads to convex optimization*
>
> We agree this would be interesting to try. A similar approach was proposed by Zafar et al. (2019), but they explicitly leave this optimization with respect to fairness definitions with non-linear statistics (like Predictive Parity) for future work. By using our $c$-fixed constraints, which construct linear constraints for linear-fractional statistics, our method directly solves this stated limitation in their work. We opted not to go this route, as the goal of our framework is to help mitigate bias in non-linear models optimized through automatic differentiation.
>
> **Questions**
>
> *The use of Partition Fairness*
>
> Our notation for sensitive values is different from most papers, as we allow for an arbitrary vector of sensitive features. Hence, we use the term ‘partition fairness’ to emphasize that the fairness definitions in our notation are only equivalent to the popular statistical fairness definitions when there is only a *single categorical* sensitive variable, i.e. one that poses a partition over the data. In our experience, the use of the term ‘group fairness’ in prior work is often vague and could also refer to equality along *multiple categorical sensitive variables*, e.g. all gender groups are treated equally and all ethnicity groups are treated equally. However, as discussed in Remark 2 (and Appendix B.2), enforcing equality for every axis of discrimination separately, neglects biases towards intersections of groups (e.g. ‘black women’ and ‘white women’) (Buolamwini et al., 2018). By formally stating the assumptions that come with partition fairness in our notation, we can clearly remark on this limitation (which is very common in the state-of-the-art).
>
> *Continuous sensitive variables*
>
> Indeed, linear-fractional statistics only measure linear effects of the sensitive variables on the numerator and denominator. For demographic parity (with the positive rate as its statistic), this is equivalent to requiring zero covariance/correlation between the sensitive variables and the output score. We show that this notation is exactly equivalent to traditional fairness metrics for a single, categorical attribute (partition fairness), but fails to capture non-linear effects for continuous sensitive variables (see also Appendix B.1 that goes into more detail). A possible extension could be to use the work of Mary et al. (2019) to estimate the joint density of the sensitive variables and the numerator and denominator of the linear-fractional statistic. By mapping these variables into a kernel space, that density becomes tractable to compute.
>
>
> **References**
>
> Buolamwini J., Gebru T., (2018). Gender Shades: Intersectional Accuracy Disparities in Commercial Gender Classification. Proceedings of the 1st Conference on Fairness, Accountability and Transparency, PMLR 81:77-91
>
> Mary, J., Calauzenes, C., & El Karoui, N. (2019, May). Fairness-aware learning for continuous attributes and treatments. In International Conference on Machine Learning (pp. 4382-4391). PMLR.
>
> Zafar, M. B., Valera, I., Gomez-Rodriguez, M., & Gummadi, K. P. (2019). Fairness constraints: A flexible approach for fair classification. The Journal of Machine Learning Research, 20(1), 2737-2778.

---

> > ### Comment · Reviewer_eBDd · 2023-11-20
> >
> > I thank the reviewers for their comments. I would like to retain my score.

---

### Official Review · Reviewer_ALkR · 2023-10-31

**Soundness:** 4 excellent
**Presentation:** 4 excellent
**Contribution:** 3 good
**Rating:** 8
**Confidence:** 4

**Summary:**

This paper presents a tool that implemented generalized group fairness metrics that can be used in automatic differentiation libraries. The fairness metrics are expressed as a linear-fractional statistic, which can be further represented as a smoothed regularization term. The authors also present an alternative projection method that penalize the divergence between models. Through extensive experiments, the authors shows the framework is lucrative for the optimization of fairness constraints.

**Strengths:**

- The presented SmoothMax regularization terms are elegant and provide expressive representations for widely applied group fairness metrics.
- The methods can be combined with automatic differentiation tools, such as PyTorch.
- The methods can be naturally applied with multiple axes of sensitive attributes, allowing wider applications.

**Weaknesses:**

- The method still applies relaxed fairness metrics, rather than the exact metrics as the regularization terms.
- A superior learning objective should be a minimax game with the optimization of the $\lambda$-player. As far as my understanding, the authors use fixed $\lambda$ values as a hyper-parameter and run grid search to get the optimal results.

**Questions:**

1. If the model $h$ is not a probabilistic classifier, then Equation (4) is no longer differentiable. Can the framework still be useful?
2. How is the projection $f^\star$ initialized in the constrained optimization problem of Equation (5)?

---

> ### Author Response · Authors · 2023-11-15
>
> We thank the Reviewer for the time they invested into reviewing the paper and for providing insightful comments. Of course, we highly appreciate their positive evaluation of our paper.
>
> **Weaknesses**
>
> *On relaxed fairness notions*
>
> Remark 3 in our paper fully agrees that, because we assume classifiers to be probabilistic in Section 3, the fairrets only use relaxed versions of the discrepancy between statistics. Yet, as we argue in our response to your Question 1, this weakness is not necessarily a problem in practice, where probabilistic classifiers are common. Moreover, as discussed in Appendix B.3, the hardness of the classification (and thus the fidelity of the fairness metric) can be traded-off with the quality of the fairret’s gradient by scaling the logits (Padh et al. 2021).
>
> *On finding $\lambda$*
>
> This is a great idea that would fit well in the envisioned framework. We share the hypothesis that a smarter $\lambda$ that evolves during training would indeed lead to better results. In fact, we noticed that setting $\lambda = 0$ for the first 20 epochs led to significantly more stable convergence (see Appendix D.2). We would only add that this illustrates the value of our modular approach to regularizers: the impact of the fairness adjustment is easily controlled.
>
> **Questions**
>
> 1. To clarify, we refer to probabilistic classifiers as classifiers that provide a distribution of outputs to sample decisions from, in response to an input feature vector. Such classifiers have become standard in modern machine learning setups, which typically use automatic differentiation libraries. Hard classifiers, with their discontinuity along the decision boundary, are unfit for this paradigm. Hence, they fall outside the target scope of our framework. Please see Appendix B.3 for a detailed discussion on the topic. However, do note that all results in Section 2 *are* directly applicable to these non-probabilistic classifiers. In particular, future work in non-probabilistic classifiers could use Proposition 1 to obtain general, linear fairness constraints.
>
> 2. We did not extensively evaluate the use of different initializations for the projection $f^*$. To make the implementation as simple as possible, we maximally benefit from the `cvxpy` library to handle the optimization. There, we saw some benefit from using a warm start (https://www.cvxpy.org/tutorial/advanced/index.html#warm-start), i.e. the previous optimization parameters are used to initialize the next optimization’s parameters. We hypothesize this is because the dual variables in the convex optimization, i.e. those related to the fairness constraints, are expected to be similar from one batch to the next, since these dual variables represent the imbalance in statistics between groups.
>
> **References**
>
> Padh, K., Antognini, D., Lejal-Glaude, E., Faltings, B., & Musat, C. (2021, December). Addressing fairness in classification with a model-agnostic multi-objective algorithm. In Uncertainty in artificial intelligence (pp. 600-609). PMLR.

---

> > ### Comment · Reviewer_ALkR · 2023-11-15
> >
> > I acknowledge that I have read the authors' response as well as the comments with other reviewers.

---

### Meta-Review · Area_Chair_mCfz · 2023-12-06

**Metareview:**

The paper introduces a new framework called "fairret" for enhancing fairness in machine learning. This framework, compatible with automatic differentiation libraries, quantifies bias using modular objectives. Additionally, a PyTorch implementation of the fairret framework is included as part of this contribution.

As the authors said, people in the field focus too much on the trade-offs and providing theoretical guarantees. There is a strong need for developing tools the practical use. The theoretical contribution of this paper is a little weak but I think that is not the main focus of this paper.

**Justification For Why Not Higher Score:**

The theoretical contribution is weak, although that is the main focus of this paper. This paper is positioned as a contribution to practical tool/software/app for building fair machine learning models.

**Justification For Why Not Lower Score:**

This paper makes nice contribution to practical tool/software/app for building fair machine learning models.

---

### Decision · Program_Chairs · 2024-01-16

Accept (poster)